# Interplay between hypoxia and androgen controls a metabolic switch conferring resistance to androgen/AR-targeted therapy

Hao Geng[1], Changhui Xue[1], Janet Mendonca[2], Xiao-Xin Sun[3], Qiong Liu[1], Patrick N. Reardon [4], Yingxiao Chen[3], Kendrick Qian[1], Vivian Hua[1], Alice Chen[1], Freddy Pan[1], Julia Yuan[1], Sang Dang[1], Tomasz M. Beer [1,5], Mu-Shui Dai[3], Sushant K. Kachhap[2] & David Z. Qian[1,5]

Despite recent advances, the efficacy of androgen/androgen receptor (AR)-targeted therapy remains limited for many patients with metastatic prostate cancer. This is in part because prostate cancers adaptively switch to the androgen/AR-independent pathway for survival and growth, thereby conferring therapy resistance. Tumor hypoxia is considered as a major cause of treatment resistance. However, the exact mechanism is largely unclear. Here we report that chronic-androgen deprivation therapy (ADT) in the condition of hypoxia induces adaptive androgen/AR-independence, and therefore confers resistance to androgen/AR-targeted therapy, e.g., enzalutamide. Mechanistically, this is mediated by glucose-6-phosphate isomerase (GPI), which is transcriptionally repressed by AR in hypoxia, but restored and increased by AR inhibition. In turn, GPI maintains glucose metabolism and energy homeostasis in hypoxia by redirecting the glucose flux from androgen/AR-dependent pentose phosphate pathway (PPP) to hypoxia-induced glycolysis pathway, thereby reducing the growth inhibitory effect of enzalutamide. Inhibiting GPI overcomes the therapy resistance in hypoxia in vitro and increases enzalutamide efficacy in vivo.

---

[1] OHSU Knight Cancer Institute, Prostate Cancer Program, Oregon Health & Science University, 3181 SW Sam Jackson Park Road, Portland, OR 97239, USA. [2] Johns Hopkins Kimmel Cancer Center, 401 N Broadway, Baltimore, MD 21287, USA. [3] Department of Medical Genetics, Oregon Health & Science University, 3181 SW Sam Jackson Park Road, Portland, OR 97239, USA. [4] NMR Core facility, Oregon State University, Corvallis, OR 97331, USA. [5] Division of Hematology & Medical Oncology, Oregon Health & Science University, 3181 SW Sam Jackson Park Road, Portland, OR 97239, USA. These authors contributed equally: Hao Geng, Changhui Xue, Janet Mendonca. Correspondence and requests for materials should be addressed to D.Z.Q. (email: qianzh@ohsu.edu)

Primary prostate cancer is treated by surgery and local radiation. The overwhelmingly oncogenic role of androgen/ AR-pathway in supporting the growth and survival of metastatic prostate cancer dictates that androgen/AR-targeted therapy is the best treatment option for patients with the metastatic disease[1,2]. Androgen deprivation therapy (ADT) or castration has been the mainstay treatment because it depletes androgen ligands that activate AR. As a result, androgen/AR-dependent gene expression and metastatic tumor growth are inhibited[3]. Although initially successful, ADT universally fails after 2–3 years, with the emergence of castration-resistant prostate cancer (CRPC), killing ~27,000 men every year in the US[4]. Recently, more effective treatments blocking the androgen/AR pathway have been developed. They include abiraterone to block androgen biosynthesis[5], enzalutamide to block AR activation[6], plus AR activity/stability-targeting agents, e.g., BET bromodomain inhibitor (JQ1)[7], dimethylcurcumin (ASC-J9)[8], and DNA sequence-specific polyamides[9]. Abiraterone and enzalutamide are FDA approved. However, the efficacy of these new treatments is still limited by insensitivity or emergence of resistance[10–12].

Recent preclinical and clinical studies have shown that in response to successful androgen/AR blockade, AR-positive prostate cancers may switch to androgen/AR-independent pathways for survival and growth, thereby conferring therapy resistance and enabling disease progression[13,14]. Some of the genes/pathways implicated in conferring resistance are similar to those underpinning the development of AR-negative or neuroendocrine prostate cancer, further suggesting the AR-independent nature. Despite the extensive effort in understanding the mechanism of resistance, most studies have not directly considered the effect of tumor hypoxia or low oxygen. The current approach is primarily based on in vitro in non-hypoxic conditions or in vivo where oxygen concentration is heterogenic and hypoxic effects can be masked by non-hypoxic effects.

Hypoxia is a pathological hallmark of solid tumors. For decades, tumor hypoxia has been considered a major culprit in treatment resistance and subsequent progression of lethal disease[15], including metastatic prostate cancer[16–18]. This is because hypoxia activates a diverse group of genes and corresponding pathways that support stress adaptation and survival[19,20]. Many of the hypoxia-response genes are oncogenic, capable of overriding the growth inhibitory activities of therapies[21]. Hypoxic cancer cells are more likely to survive therapies and grow compared to their normoxic counterparts[22]. Hypoxia-inducible factor (HIF) is induced and activated by hypoxia and is primarily responsible for upregulating these genes by serving as the master transcription factor[21]. HIF is a heterodimer of HIFα/HIFβ proteins, and has two major isoforms, HIF1 (HIF1α/HIF1β) and HIF2 (HIF2α/HIF1β)[15].

Emerging clinical evidence suggests that hypoxia and HIF1 may have a significant role in the progression of metastatic prostate cancer, CRPC development, and treatment resistance.[23–34]. Despite the intriguing finding, the exact molecular mechanism by which hypoxia drives insensitivity and resistance to androgen/AR-targeted therapy remains unclear. The molecular relationship between the two important oncogenic axes (androgen & hypoxia) remains largely unexplored (Supplementary Fig. 1).

## Results
### Chronic ADT in hypoxia induces enzalutamide resistance.
Resistance to androgen/AR-targeted therapies is driven by multiple mechanisms and arises following chronic-treatment in vitro and/or in vivo[14]. In order to determine the hypoxia-induced cause, we took an in vitro selection approach by chronically treating the androgen-dependent LNCaP and LAPC4 cells with

multiple rounds of ADT in hypoxia (1% $O_2$) (Fig. 1a). Cells treated in normoxia (20% $O_2$) or cultured in synthetic androgen (R1881, 1 nM) were used as controls. After ~15 rounds, we labeled the four resulted cell clones based on the selective pressure, e.g., ADT-in-hypoxia-selected (AdtHs). We then tested their sensitivities in normoxia and hypoxia to androgen/AR-targeted disruptions with enzalutamide or AR-siRNA (Supplementary Fig. 2a, b). In 20% $O_2$, we found that all cells displayed similar sensitivity to the growth inhibitory effect of enzalutamide (Fig. 1b) or AR-siRNA (Fig. 1c). In 1% $O_2$, however, the AdtHs cells, derived from chronic-ADT-in-hypoxia, were less sensitive to the treatment (Fig. 1b, c). Cell cycle analyses revealed that enzalutamide or AR-siRNA effectively induced cell cycle arrest (G1) and cell death (sub-G0/G1) in AdtHs cells in 20% $O_2$ (Fig. 1d). However, the growth inhibitory effect was lost under the hypoxic condition (Fig. 1d). In contrast, the treatment was effective in both oxygen conditions in other clones, e.g., AdtNs (ADT-in-normoxia-selected) (Fig. 1d). The gain of growth/survival advantage by AdtHs cells in hypoxia was independent of androgen/AR, because the activity of androgen/AR signaling, as seen by AR-target gene expressions, was effectively inhibited in both oxygen conditions (Fig. 1e). Similar results were found in the LAPC4 (Fig. 1f). To determine the sensitivity to enzalutamide in vivo, we established xenografts with AdtHs and AdtNs cells. We found that enzalutamide was significantly more effective in inhibiting the AdtNs tumors than the AdtHs tumors (Fig. 1g). Taken together, these data suggest that chronic-therapy (ADT)-in-hypoxia may select prostate cancer cells capable of displaying androgen/AR-pathway independence and therapy resistance in hypoxia in vitro and in xenograft tumors in vivo.

### Androgen attenuates the hypoxia-induced gene upregulation.
Next, we explored the mechanism by which chronic-ADT may select therapy resistance in hypoxia. Since hypoxia and androgen exert biological effects primarily via regulation of gene expression, we performed cDNA microarray with parental LNCaP cells that were cultured for 24 h in androgen-free ADT (basal), hypoxia (1% $O_2$), androgen (1 nM R1881), and hypoxia + androgen (Combo) conditions. Then, we used false-discovery adjusted t-test to determine genes that were significantly altered by hypoxia (Fig. 2a), androgen (Fig. 2b), and the combination (Fig. 2c). Next, we compared the expression of hypoxia-response genes in hypoxia vs. in the combination (hypoxia + androgen) condition. We found that a subset of the hypoxia-response genes expressed differently with or without androgen (Fig. 2d), and androgen was able to enhance or attenuate the hypoxic effect (Fig. 2g). With the same approach, we identified the subset of androgen target genes that were significantly modified by hypoxia (Fig. 2e), and the subset of combination target genes that were significantly different from androgen and hypoxia conditions alone (Fig. 2f).

Hypoxia-response genes may support cancer cell survival and growth[15]. Thus, the androgen-attenuated hypoxia-response genes in the shaded area of Fig. 2g were particularly interesting to us. It suggested that in response to ADT in hypoxia, these genes can be restored and increased, and some of which may in turn confer androgen/AR-independent prostate cancer and thus therapy resistance. Supporting this possibility, we found that the negative effect of androgen on hypoxia was also significant at the pathway level. Gene set enrichment analysis (GSEA) showed that androgen significantly attenuated the ability for hypoxia to upregulate subsets of cancer hallmark pathways (Supplementary Fig. 3a). Conversely, ADT-in-hypoxia was more capable of increasing subsets of cancer hallmark pathways than ADT-in-normoxia (Supplementary Fig. 3b). Three pathways and the associated genes were consistently attenuated by androgen but

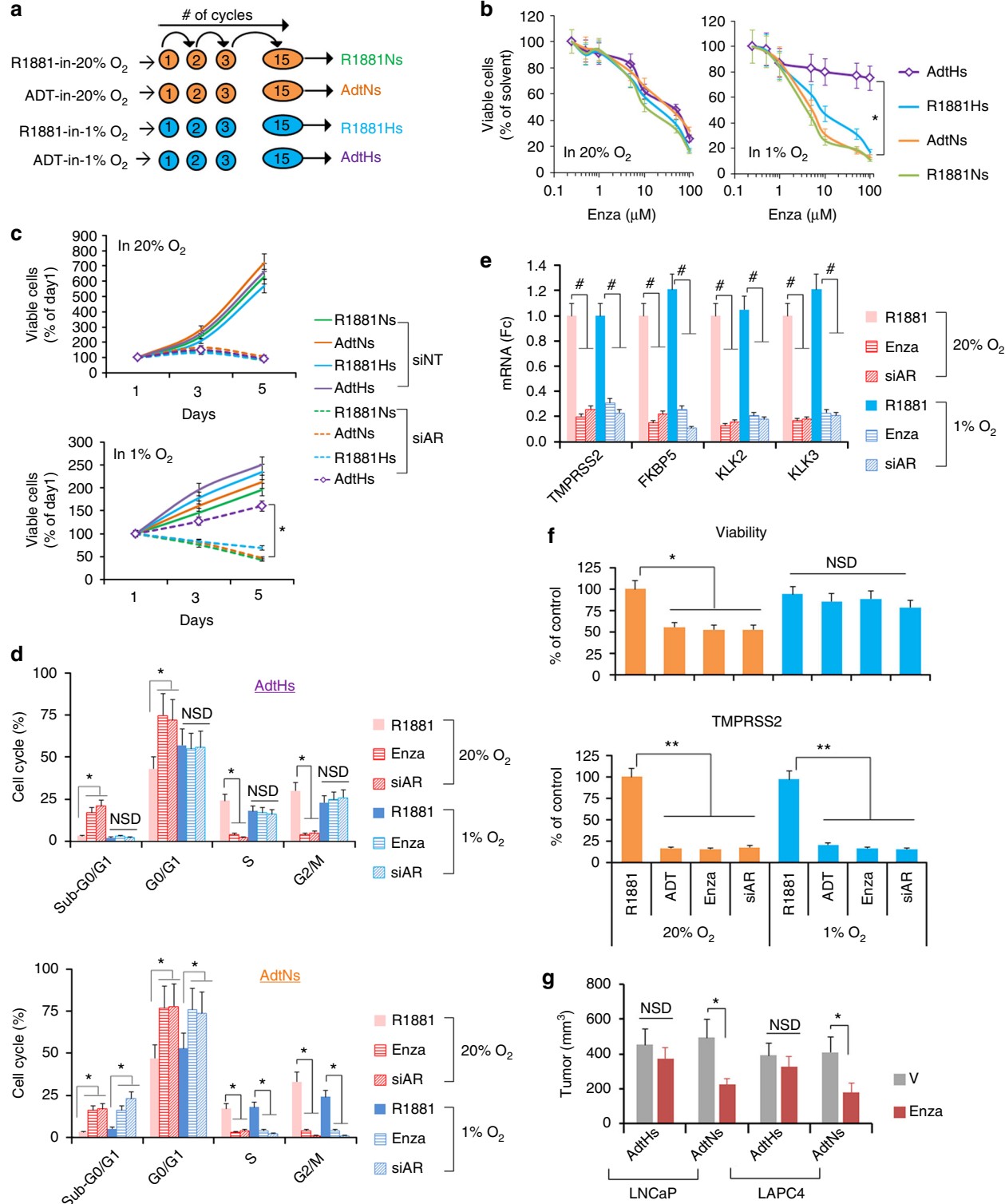

increased by ADT in hypoxia—estrogen response, mTORC1, and unfolded protein response (UPR) (Supplementary Fig. 3c–e).

To further explore, we analyzed the cDNA arrays collected from the treatment resistant LNCaP-AdtHs and LAPC4-AdtHs cells under same conditions as parental LNCaP. We found that these cells shared a common group of androgen-attenuated hypoxia-response genes (Fig. 2h, Supplementary Fig. 3f). A subset of these genes was subsequently confirmed with qRT-PCR. These genes were significantly increased by androgen/AR-targeted treatment + hypoxia, but not by the treatment or hypoxia alone

(Fig. 2i, Supplementary Fig. 3g). Interestingly, the level of increase by treatment + hypoxia was significantly higher in AdtHs than in parental cells (Fig. 2i), suggesting (i) preferential selection by chronic-ADT + hypoxia, and (ii) that these changes were not just the result of resistance, but potential causes.

To understand the clinical involvement of the androgen-attenuated hypoxia-response genes in Fig. 2h and Supplementary Fig. 3f, we performed a survival analysis with the mRNA level of each gene in the prostate cancer TCGA dataset (Fig. 2j). We find that five of these genes (*IRF2BP2, EDARADD, GPI, HMOX1,*

**Fig. 1** Chronic-ADT in hypoxia induces resistance to targeted disruption of androgen/AR-axis. **a** Schematic representation of repeated-ADT treatment of LNCaP and LAPC4 cells in hypoxia (1% $O_2$) or normoxia (20% $O_2$). Cells were cultured in charcoal-stripped serum (CSS) media with or without 1 nM synthetic androgen R1881 (R1881 or ADT). Each round/cycle of treatment lasted 48 h, and the survived cells were recovered and re-plated for the same treatment in the following cycle. **b** Dose-dependent sensitivity of LNCaP clones to enzalutamide. Cells derived from **a** were cultured with CSS + R1881, and treated by increasing doses of enzalutamide in normoxia or hypoxia for 96 h. Afterwards, cell growth and viability were determined based on viable cells quantitated with SYTO 60 fluorescence staining and imaging. The AdtHs is represented by purple line. All values were mean ± s.d. relative to solvent-treated controls, $n = 3$. *$P < 0.01$ two-sided $t$-test. **c** Time-dependent sensitivity of LNCaP clones to AR-siRNA. Cells derived from **a** were transfected with a cocktail of siRNA constructs silencing AR (siAR) or non-target control (siNT), and then cultured in media with CSS + R1881 in normoxia or hypoxia for 96 h. Viable cells were quantitated with SYTO 60 every 48 h. The AdtHs treated by siNT or siAR is represented by solid purple or dotted purple line, respectively. All values were mean ± s.d. relative to the beginning of the siRNA at day 1, $n = 3$ *$P < 0.01$ two-sided $t$-test. **d** LNCaP-AdtHs or AdtNs cells, cultured in media with CSS + R1881, were treated with negative control (R1881), 10 uM enzalutamide (Enza), or siAR for 48 h in normoxia or hypoxia. The cell cycle distribution was determined by flow cytometry, mean ± s.d. $n = 3$. *$P < 0.05$, NSD (no significant difference), two-sided $t$-test. **e** LNCaP-AdtHs cells were treated as in **d**. The AR-target gene expression was determined by qRT-PCR, mean ± s.d. relative to R1881 in normoxia, $n = 3$, #$P < 0.01$ two-sided $t$-test. **f** The growth/viability and *TMPRSS2* gene expression of LAPC4-AdtHs cells, cultured with CSS + R1881, after being treated by negative control (R1881), ADT, 10 μM enzalutamide or siAR for 96 h in normoxia or hypoxia. All values are mean ± s.d. $n = 3$, *$P < 0.05$, **$P < 0.01$, NSD, two-sided $t$-test. **g** The antitumor activity of enzalutamide in nude mice against subcutaneous xenografts established with AdtHs or AdtNs cells. Equal numbers of indicated cells (~5 million) were injected into male nude mice. Enzalutamide (Enza) or vehicle (V) treatment started when the tumors were ~100 mm³. All values are mean ± s.d. $n = 6$, *$P < 0.05$, two-sided $t$-test

*KIF3C*) were over-expressed in patient samples (Fig. 2k), and the overexpression was significantly associated with prostate cancer relapse (the onset of metastatic disease) as individual (Fig. 2j) and as a group (Fig. 2l). In the molecular signature database (MSigDB), these five genes are involved with oncogenic signaling, e.g., androgen response (*IRF2BP2*), hypoxia (*GPI, HMOX1*), mTORC1 (*GPI*), mitotic checkpoint (*KIF3C*), and poor prognosis mutant drivers of prostate cancer, e.g., TP53 (*HMOX1*) and PTEN/AKT (*EDARADD*).

**Androgen attenuates hypoxia-induced GPI upregulation via AR.** Among the potentially clinical-relevant androgen-attenuated hypoxia-response genes, we were specifically interested in glucose-6-phospate isomerase (GPI), which is the enzyme directing glucose metabolism to the glycolysis pathway—an essential cellular adaptation and survival response to hypoxia[35–37]. In the prostate cancer TCGA dataset, the overexpression of GPI occurred to ~4% of patients (Fig. 2k), and was significantly associated with shorter relapse-free survival (Fig. 2j, Supplementary Fig. 4a). In additional clinical prostate cancer datasets, the overexpression of GPI occurs more frequently in androgen/AR-negative neuroendocrine prostate cancer (Supplementary Fig. 4b), suggesting a negative association with the androgen/AR-axis.

We determined the expression of GPI in all the parental and therapy-selected prostate cancer clones. In basal (ADT) condition, hypoxia significantly increased the expression of GPI in all clones; however, 1 nM R1881 attenuated the hypoxia-induced upregulation, which can be restored by enzalutamide (Fig. 3a, Supplementary Fig. 5a). Interestingly, the increase of GPI by hypoxia + ADT/enzalutamide was most robust in the therapy-resistant AdtHs (Fig. 3a, Supplementary Fig. 5a). Consistent with the mRNA, we found that enzalutamide treatment in hypoxia increased the enzymatic activity of GPI in all clones, and the magnitude of increase was most significant in the resistant AdtHs (Fig. 3b, Supplementary Fig. 5b), suggesting a potential selection advantage. In AdtHs cells, we also used western blots to confirm the attenuation of hypoxia-induced GPI by androgen (Fig. 3c), and the restoration or increase by enzalutamide (Fig. 3d).

To determine the roles of HIF and AR in the regulation of GPI, we inhibited HIF1α, HIF2α, or AR with previously established siRNA constructs. We found that HIF1α, not HIF2α, was primarily responsible for the hypoxic upregulation of GPI (Fig. 3e, Supplementary Fig. 5c). Importantly, when AR was silenced by siRNA, the ability for androgen to attenuate the hypoxic upregulation was significantly negated (Fig. 3e, Supplementary

Fig. 5c), suggesting that the attenuation was mediated through AR. To further test the role of AR, we determined the level of GPI in AR-negative DU145 cells, and DU145 with stable AR transfection (a gift from Dr. John Isaacs at Johns Hopkins University). Previously, the over-expressed AR has been shown to have androgen-independent and constitutive activities[38]. We found that hypoxia significantly increased GPI expression in the AR-negative DU145+Ev cells; in contrast, it was significantly attenuated in AR-positive DU145+AR cells (Fig. 3f, Supplementary Fig. 5d). Further, AR-siRNA restored/increased the GPI expression in hypoxia in DU145+AR cells, while having no effect in DU145+Ev cells (Fig. 3g & Supplementary Fig. 5d).

AR exerts the transcriptional regulation by binding to its target genes. We found that an androgen response/AR-DNA binding element half-site (AREhs), 5′-AGAACA-3′, was adjacent to a hypoxia-response/HIF-DNA binding element (HRE), 5′-ACGTG-3′, in the GPI promoter (Fig. 3h and Supplementary Fig. 6). Therefore, we used site-directed mutagenesis to eliminate the AREhs or the HRE in a GPI promoter-driven luciferase reporter gene construct (Fig. 3h). Luciferase reporter assay in the presence of AR (siNT) showed that hypoxia increased luciferase activity in the wild-type (wt) construct, and androgen attenuated the hypoxic increase, in contrast, silencing AR with siRNA (siAR) negated the attenuation (Fig. 3i). In the promoter without AREhs (ΔAREhs), however, androgen was unable to attenuate the hypoxic upregulation, and AR-siRNA also had no effect (Fig. 3i). We further found similar AR and AREhs-dependent effects in AR-over-expressed DU145 cells (DU145+AR), but not in the AR-negative counterparts (DU145+Ev) (Fig. 3j).

**AR interferes with HIF1-mediated GPI transcription.** Our microarray results suggested that the negative effect of androgen/AR occurred to only a subset of hypoxia-response genes. Unlike GPI, for example, the hypoxic upregulation of enolase 1 (ENO1) was not attenuated by androgen or restored by enzalutamide or AR-associated BRD4 targeting agent JQ1[7] (Fig. 4a). Unlike GPI, interestingly, there is no androgen response/AR-DNA binding element adjacent to the functionally verified HRE within the ENO1 promoter (Supplementary Fig. 6).

Chromatin immunoprecipitation (ChIP) analyses showed that androgen stimulation in normoxia and hypoxia increased AR binding (Fig. 4b), and enriched the transcriptional-unfavorable chromatin modification, 2me-H3K9 (Fig. 4c) at the GPI promoter, but not at the ENO1 promoter (Fig. 4b, c). Accordingly, targeting androgen/AR with enzalutamide or JQ1

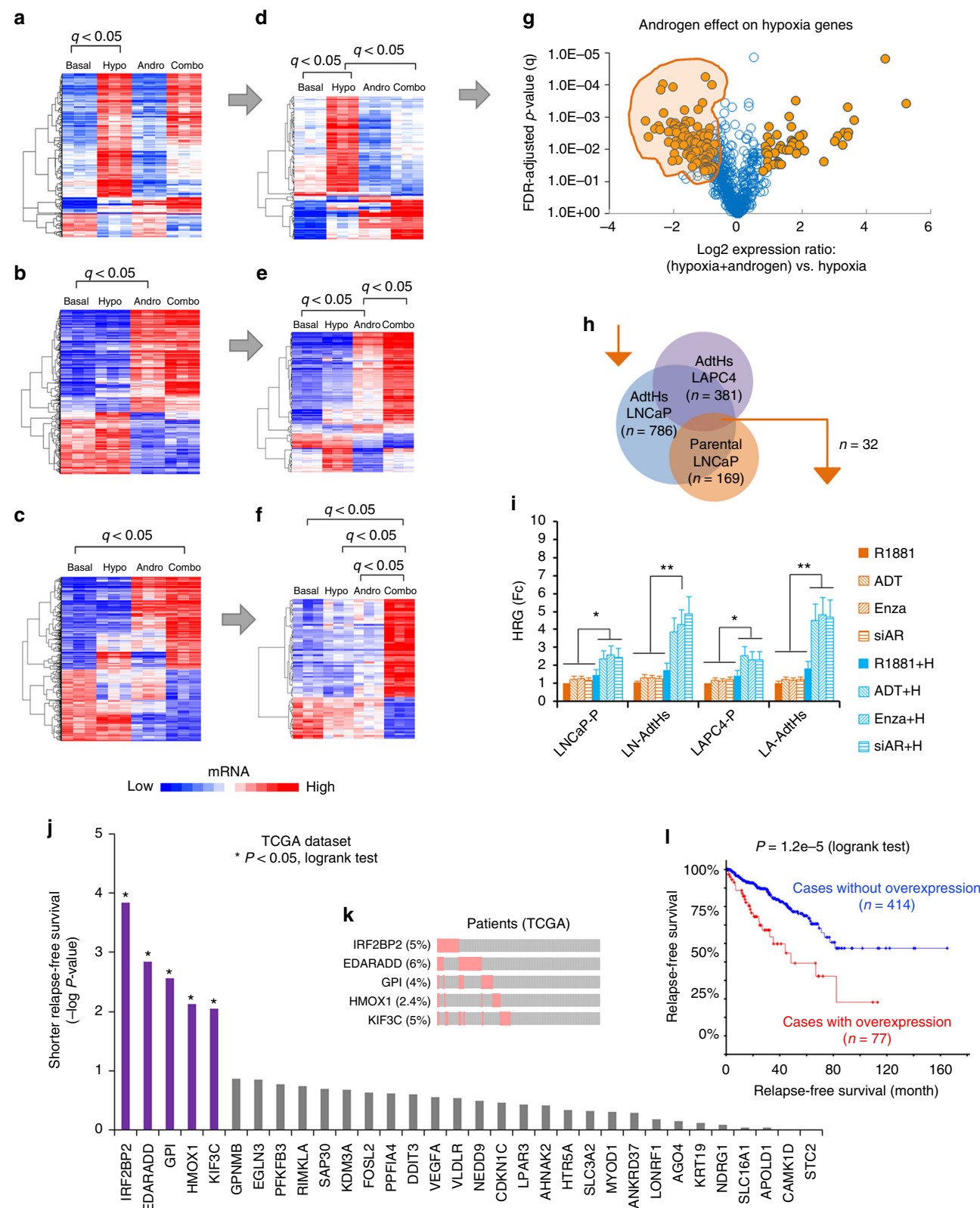

reversed the effect of androgen at the GPI promoter (Fig. 4b, c), and had no effect at the ENO1 promoter (Fig. 4b, c). ChIP also showed that hypoxia alone increased the enrichment at both promoters for HIF1α (Fig. 4d), histone acetylation, ace-H3K9 (Fig. 4e), histone aectylase/transcription co-factor p300 (Fig. 4f), and RNA polymerase II (Fig. 4g), all of which are in favor of transcriptional activation or upregulation. At the GPI promoter, however, adding androgen to hypoxia (combo) significantly reduced these transcriptionally favorable markers (Fig. 4d–g). On the other hand, enzalutamide or JQ1 inhibited the androgenic effect (Fig. 4d–g). In contrast, androgen or targeted therapies did not change the promoter of ENO1 (Fig. 4d–g). In summary, these

**Fig. 2** Androgen/AR attenuates a subset of hypoxia-induced gene expression. LNCaP cells were cultured in basal (CSS media), hypoxia (hypo, 1% $O_2$), androgen (andro, 1 nM R1881), or the combination (combo, hypoxia + androgen) conditions ($n = 3$) for 24 h. Afterwards, gene expression profiles were determined with cDNA microarray, and compared by false-discovery (FDR) adjusted $t$-test. The significance was determined as log2 > +0.75 or <−0.75, FDR-adjusted $P$-value ($q$) < 0.05. **a–c** Heatmaps of genes that were significantly altered by hypoxia (**a**), androgen (**b**), or hypoxia + androgen (**c**). **d–f** Heatmaps of hypoxia-response genes that were significantly altered by androgen in the combination condition (**d**), androgen-response genes that were significantly altered by hypoxia (**e**), and combination-response genes that were significantly different from the single condition (**f**). **g** Volcano plot analysis to identify the hypoxia-response genes that were most significantly changed by androgen. The genes in the shaded area were significantly attenuated as androgen-attenuated hypoxia-response genes, log2 (combo vs. hypoxia) < −0.75, $q$ < 0.05, $n = 3$, FDR-adjusted $t$-test. **h** Venn diagram of androgen-attenuated hypoxia-response genes in parental LNCaP (**g**), LNCaP-AdtHs, and LAPC4-AdtHs. The n represents the number of genes identified. **i** The parental (P) or AdtHs cells from LNCaP (LN) and LAPC4 (LA), cultured in CSS + 1 nM R1881, were treated with negative control (R1881), androgen/AR-blockade (ADT, 10 μM enzalutamide, or siAR), hypoxia (H), or androgen/AR-blockade + hypoxia for 48 h. The expressions of androgen-attenuated hypoxia-response genes identified in **h** were determined by qRT-PCR. The negative effect of androgen/AR on hypoxia was confirmed in 15 genes (Supplementary Fig. 3g), values are group mean ± s.d. relative to negative control in normoxia, $n = 15$, *$P$ < 0.05, **$P$ < 0.01 paired $t$-test. **j** The 32-gene geneset identified in **h** was used to find the association between gene expression and disease-free survival in prostate cancer TCGA dataset on cBioPortal website, *$P$ < 0.05, log-rank test, $n = 491$. **k** The top-5 significant association in **j** is shown with the % of mRNA alteration in patients and the type of alteration, red is overexpression, gray is no change. **l** Kaplan–Meier analysis of disease-free survival in relationship to mRNA level alterations in prostate cancer TCGA dataset stratified with the top five genes in **j–k** $P = 1.2e−5$, log-rank test, $n = 491$

data suggest a molecular model of negative crosstalk between androgen and hypoxia at the GPI promoter, whereby AR and the HRE-adjacent AREhs interfere with HIF1/HRE-mediated gene upregulation in hypoxia.

**ADT or enzalutamide in hypoxia rewires glucose flux via GPI.** Upon cellular uptake and phosphorylation, the glucose metabolite, glucose-6-phosphate (G6P), is either metabolized by GPI to enter the glycolysis pathway, or by glucose-6-phosphate dehydrogenase (G6PD) to enter the biosynthetic pentose phosphate pathway (PPP) (Fig. 5a). In prostate cancers, androgen/AR promotes glucose metabolism in normoxia. It does so via directly increasing the expression of enzymes for glucose uptake (Glut1), conversion to G6P (HK1), and flux through PPP (G6PD and NUDT9)[39,40] (Fig. 5a). However, it is unclear how glucose metabolism is regulated in response to androgen/AR-targeted therapies in normoxia vs. hypoxia. We traced the metabolic flux of [U-$^{13}$C]-labeled-glucose in LNCaP-AdtHs cells cultured with 1 nM R1881 (control), ADT, hypoxia, or ADT + hypoxia. In normoxia, ADT reduced the glucose metabolism, including glucose uptake and conversion to G6P (Fig. 5b), metabolites along the PPP leading to nucleotide biosynthesis (Fig. 5c), and metabolites along the glycolysis pathway leading to lactate and the mitochondrial TCA cycle (Fig. 5d). In hypoxia, ADT was still able to reduce the PPP (Fig. 5c). However, the glucose uptake and conversion to G6P were maintained (Fig. 5b), and metabolites along the cytosolic/non-mitochondrial glycolysis pathway leading to the production of lactate and alanine were all increased (Fig. 5d). This switch from PPP to the cytosolic glycolysis was not caused by hypoxia alone, but by the combination of ADT + hypoxia, as hypoxia alone did not reduce PPP or increase glycolysis (Fig. 5c, d). This suggested that the glucose uptake and metabolism are reduced with ADT in normoxia, but maintained with ADT in hypoxia due to the redirection or rewiring of glucose flux from PPP to glycolysis.

To understand the role of GPI in glucose metabolic rewiring above, we used siRNA to inhibit GPI in AdtHs cells (Supplementary Fig 7a, Fig. 5e). In cells with siNT, we found similar results to the glucose tracing above. Enzalutamide decreased PPP-metabolite 6-PGA (Fig. 5f), but increased GPI enzymatic activity (Fig. 5e) and its metabolite F6P (Fig. 5g) in hypoxia. In cells with GPI-siRNA, however, the basal and enzalutamide + hypoxia-induced GPI activity and F6P were all significantly reduced (Fig. 5e, g), suggesting GPI is critical to the therapy + hypoxia-induced glucose metabolism rewiring from PPP to glycolysis. In the enzalutamide + hypoxia condition, we further found that

G6P was increased (Fig. 5h), while the glucose uptake was decreased (Fig. 5i) by siGPI, reminiscent to previous reports that GPI inhibition and accumulation of G6P may inhibit glucose uptake via feedback mechanisms[41]. Taken together, these data suggest that androgen/AR-blockade in hypoxia decreases the glucose flux via PPP, but increases GPI-dependent glycolysis. Therefore, the glucose metabolism is maintained in response to therapy-in-hypoxia via redistribution and rewiring, switching from PPP to glycolysis.

To further determine the interaction between AR and GPI in the glucose pathway switch, we determined the levels of 6-PGA and F6P in DU145 (Ev) and DU145+AR cells (+AR). We found that AR overexpression promoted the PPP-metabolite 6-PGA in both oxygen conditions (Supplementary Fig. 8a), while significantly decreasing the glycolytic metabolite F6P in hypoxia (Supplementary Fig. 8b). The change of F6P by hypoxia and AR was sensitive to the GPI metabolic inhibitor 2-deoxy-glucose (2DG) (Supplementary Fig. 8b), confirming that the negative regulation of GPI expression by AR in hypoxia has functional consequences in glucose metabolism.

**GPI confers enzalutamide resistance in vitro and in vivo.** To determine whether the glucose rewiring is involved in enzalutamide resistance, we first measured the F6P level in all the LNCaP and LAPC4 clones. Consistent with the regulation of GPI mRNA and activity, the increase of F6P by enzalutamide was the highest in the resistant AdtHs cells in hypoxia (Fig. 6a). To determine whether GPI confers enzalutamide resistance, we treated the AdtHs cells with GPI-inhibiting siRNA, and measured the dose-dependent response to enzalutamide. In normoxia (20% $O_2$), cells remained sensitive to enzalutamide, and GPI inhibition did not generate additional antitumor effect (Fig. 6b). In hypoxia (1% $O_2$), cells were resistant to enzalutamide; importantly, combination of enzalutamide with GPI-siRNA significantly inhibited the resistant cells (Fig. 6b). On the other hand, this combination produced less growth inhibitory effect against the enzalutamide-sensitive cells, e.g., AdtNs, in hypoxia (Supplementary Fig 9).

To determine whether the GPI-mediated metabolic switch to glycolysis was responsible for the enzalutamide resistance in hypoxia, we measured the dose-dependent response of AdtHs cells to enzalutamide, a GPI metabolic inhibitor - 2-deoxy-glucose (2DG), or enzalutamide + 2DG. Similar to the effect of GPI-siRNA, 2DG did not significantly improve the growth inhibitory effect of enzalutamide in normoxia (Fig. 6c, Supplementary Fig. 10). In contrast, the 2DG + enzalutamide combination significantly inhibited the AdtHs cells in hypoxia (Fig. 6c,

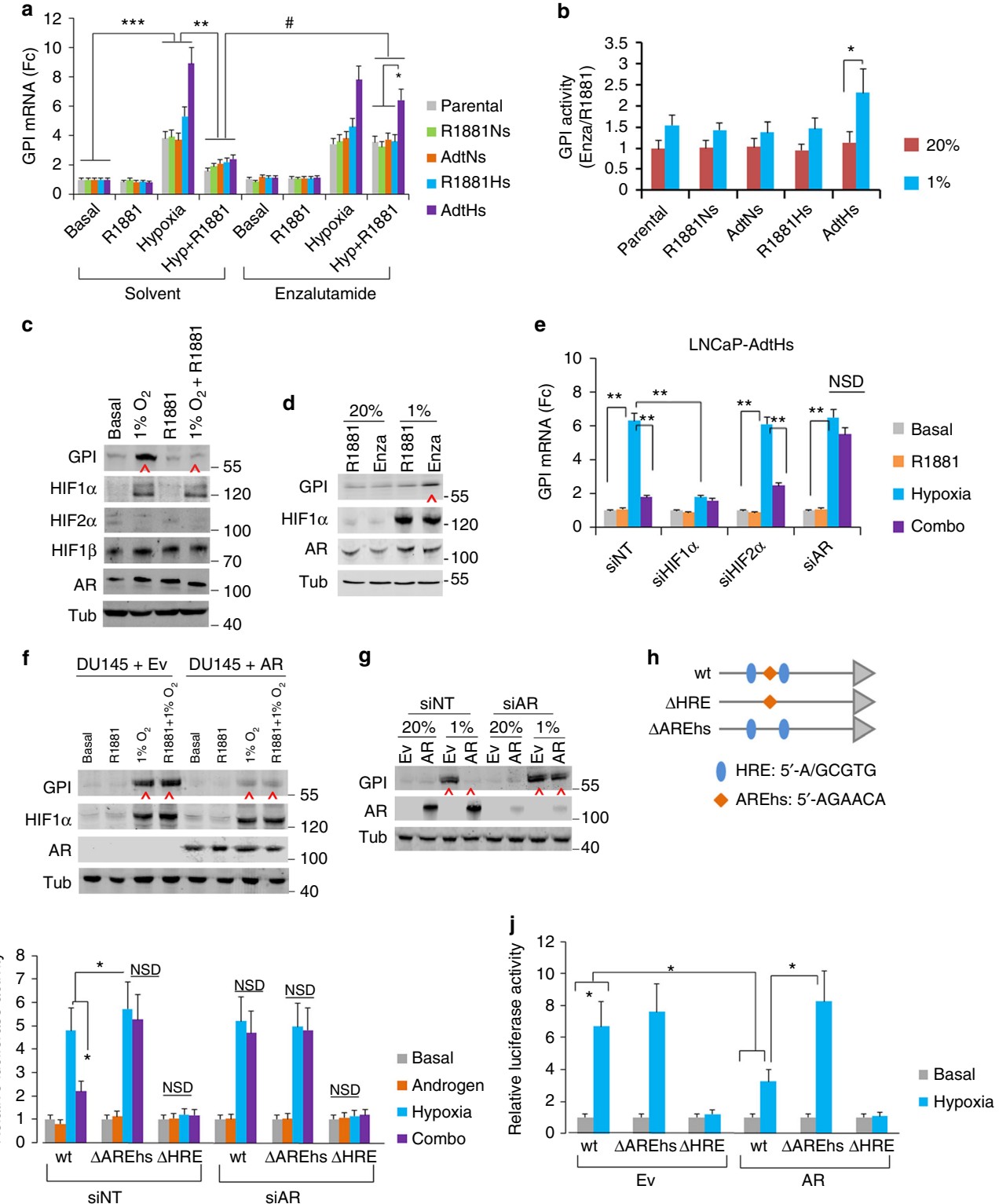

Supplementary Fig. 10). To further evaluate the enzalutamide resistance due to the glycolytic switch, we used siRNA to knockdown GPI or a GPI-downstream glycolytic enzyme, PFK1. In normoxia, AdtHs cells were sensitive to enzalutamide, and none of the siRNA further improved the growth inhibitory effect; in hypoxia, cells were resistant to enzalutamide, and targeting glycolysis with siRNA silencing GPI or PFK1 was both effective in inhibiting the resistant cells (Fig. 6d). The enzalutamide efficacy was coincided with the cellular ATP level, which was

largely maintained in normoxia, but in hypoxia enzalutamide + siNT treatment increased ATP level, while siRNA silencing the glycolytic enzymes GPI or PFK1 abolished the increase (Fig. 6e).

To determine whether GPI is responsible for the emergence of therapy resistance in the chronic-ADT-in-hypoxia model, we performed chronic-ADT experiments similar to Fig. 1a with or without low concentration (nontoxic dose) of 2DG. At cycle 1, 5, 10, and 15, cell viability and survival were determined (Fig. 6f). In

**Fig. 3** GPI expression is oppositely regulated by AR and androgen/AR-targeted therapy in hypoxia. **a** The change of GPI mRNA in LNCaP clones in response to androgen (1 nM R1881) and enzalutamide (10 μM) in normoxia or hypoxia. All cells were treated with the indicated conditions for 48 h; the basal condition was CSS media without androgen. GPI mRNA was determined by qRT-PCR. All values were mean ± s.d. relative to the basal-solvent condition, $n$ = 3, *, #$P$ < 0.05, **, ***$P$ < 0.01, repeated measures ANOVA. **b** LNCaP clones were cultured with CSS + 1 nM R1881, and treated by enzalutamide (Enza) or solvent control (R1881) in 20% or 1% $O_2$ for 48 h. The enzymatic activity of GPI was measured and expressed as mean ± s.d. relative to solvent control, $n$ = 3, *$P$ < 0.05, two-sided $t$-test. **c** Representative western blots with LNCaP-AdtHs cells cultured with basal (CSS media), hypoxia (1% $O_2$), androgen (1 nM R1881), or hypoxia + androgen conditions for 48 h. Tubulin (Tub) was used as loading control. **d** Representative western blots with LNCaP-AdtHs cells, cultured in CSS + 1 nM R1881 media, and treated with 10 μM enzalutamide (Enza) in normoxia (20%) or hypoxia (1%) for 48 h. **e** GPI mRNA levels in LNCaP-AdtHs cells in the presence of siRNA silencing HIF1α, HIF2α, AR, or non-targeting control (siNT). Cells were transfected with siRNA, and cultured in the indicated conditions for 72 h. All values are mean ± s.d. relative to the basal condition, $n$ = 3, **$P$ < 0.01, two-sided $t$-test and/or repeated measures ANOVA. **f** Representative western blots with DU145+Ev or DU145+AR cells treated by ADT (basal), 1 nM R1881, 1% $O_2$, or combination of hypoxia + R1881 for 48 h. **g** Representative western blots with DU145+Ev/AR cells treated by siNT or siAR for 72 h in 20% or 1% $O_2$. **h** Schematic representations of GPI promoter-driven luciferase plasmids, wild-type (wt) or mutants without the hypoxia-response elements (ΔHRE) or androgen response/AR-binding element half-site (ΔAREhs). **i** Relative luciferase activity driven by wild-type (wt) or mutant (ΔAREhs or ΔHRE) GPI promoter in LNCaP-AdtHs cells. Cells were first transfected with siRNA (siNT or siAR). Then, wt or mutant reporter plasmids were co-transfected with constitutive GFP plasmids. The transfected cells were cultured in basal, androgen, hypoxia or combination condition for 48 h. The luciferase values were determined, adjusted by GFP, and expressed as mean ± s.d. relative to basal condition, $n$ = 3, *$P$ < 0.05, NSD, two-sided $t$-test. **j** Relative luciferase activity driven by wild-type (wt) or mutant (ΔAREhs or ΔHRE) GPI promoters in DU145+Ev (Ev) or DU145+AR (AR) cells in normoxia (basal) or hypoxia condition. All values are means ± s.d. $n$ = 3, *$P$ < 0.05, two-sided $t$-test

normoxia, cells remained sensitive to the growth inhibitory activity of ADT or ADT+2DG combination (both) throughout the treatment (Fig. 6f). In hypoxia, cells gradually broke out of the growth inhibitory effect of ADT between cycles #5 and #10; but combination with 2DG (both) significantly blocked the resistance (Fig. 6f). Next, we determined whether increasing cellular GPI level may promote therapy resistance by transiently overexpressing a GPI-coding plasmid in parental LAPC4 (Supplementary Fig 7b). We found that GPI transfection significantly increased the enzymatic activity in both normoxia and hypoxia (Fig. 6g). In normoxia, the overexpression did not impact the efficacy of ADT; in hypoxia, however, GPI overexpression significantly dampened the growth inhibitory effect of ADT (Fig. 6h). This resistant effect by GPI paralleled with its ability to maintain ATP biosynthesis in hypoxia (Fig. 6i). These data suggested that the GPI-dependent glycolysis pathway is essential for adaptive response and resistance to androgen/AR-targeted therapy (e.g., ADT or enzalutamide) in hypoxia.

To determine whether GPI is involved in enzalutamide resistance in vivo, we first analyzed the LNCaP xenografts derived from enzalutamide-sensitive AdtNs cells and the resistant AdtHs cells in Fig. 1g. For each tumor, we dissected 4–5 separate frozen regions and determined the mRNA expression of TMPRSS2, GPI and ENO1 with qRT-PCR. In both AdtNs and AdtHs groups, tumors from the enzalutamide-treated mice had significantly lower TMPRSS2 expression than the vehicle-treated (Fig. 7a), suggesting the inhibition of AR. The GPI level was significantly higher in the resistant AdtHs tumors from the enzalutamide-treated mice (Fig. 7b). On the other hand, the hypoxia-response gene ENO1 level was not different between vehicle control and enzalutamide treatments (Fig. 7c). Further, we found that GPI had a significantly positive correlation with hypoxia-response gene ENO1 ($P$ < 0.001, $R^2$ = 0.5653, liner regression, $n$ = 25) only in the resistant AdtHs tumors + enzalutamide treatment (Fig. 7d), suggesting that GPI expression is positively associated with adaptive treatment response/resistance in hypoxia.

We next used the CRISPR-Cas9 approach to knockout (KO) GPI in LNCaP-AdtHs cells (Supplementary Fig. 11a). Similar to the effect of siRNA, the GPI-KO blocked the metabolic rewiring of glucose metabolism (Supplementary Fig. 11b, d), and increased the growth inhibitory activity of enzalutamide in hypoxia (Supplementary Fig. 11e). Subcutaneous xenografts of GPI knockout cells (ΔGPI) and GPI+ control cells (Ctl) were established in male nude mice. When the tumors were palpable (~100 mm³), mice were randomized and treated with vehicle (V) or anti-androgen/AR drug enzalutamide (Enza) for 7 weeks. The antitumor effect of enzalutamide was determined by comparing the tumor volume before and after the 7-week-treatment period. We found that enzalutamide treatment was only marginally effective in tumors with GPI (Fig. 7e); importantly the combination of enzalutamide with GPI-KO significantly improved the antitumor activity as compared to single agents (Fig. 7e). To further confirm the improved antitumor effect due to the combination, we established xenograft tumors with LAPC4-AdtHs cells, and treated the tumor bearing mice with vehicle, enzalutamide, 2DG, or combination. We found that the tumor growth was marginally sensitive to enzalutamide or 2DG alone, but was significantly arrested by the combination (Fig. 7f). These data suggest that the AdtHs tumors had an increased dependency to GPI and the downstream glycolysis pathway in response to enzalutamide in hypoxia, thus they were also more vulnerable to the genetic knockout of GPI or glycolysis-targeted treatment by 2DG. It also suggests that targeting GPI or glycolysis can be rationally used to overcome enzalutamide resistance induced by chronic-ADT in the hypoxic condition.

## Discussion
Resistance to ADT or androgen/AR-blockade can be mediated via multiple mechanisms. With the recent advance in targeting the androgen/AR-axis, treatment resistance due to the development of androgen/AR-independent prostate cancer is expected to become a major obstacle[42]. New mechanistic understanding of this type of resistance is needed to make the current treatment modality more effective and sustainable. Although tumor hypoxia is a pathological hallmark of clinical cancers and has long been suspected as a major contributor to treatment failure/resistance, the exact mechanism is yet to be clearly elucidated. This is in part because current molecular understanding of prostate cancer therapy resistance is derived by in vitro experiments conducted in normal/ambient 20% $O_2$, or genomic analyses of clinical tumor samples derived from tissues heterogenic in oxygenation.

By studying the hypoxia-specific adaptive response and resistance to ADT and enzalutamide, we present a molecular model in which androgen/AR-independence and therapy resistance may both arise from successful blockade of androgen/AR-axis (by ADT, enzalutamide, or siAR etc.) in hypoxia (Fig. 7g). In our

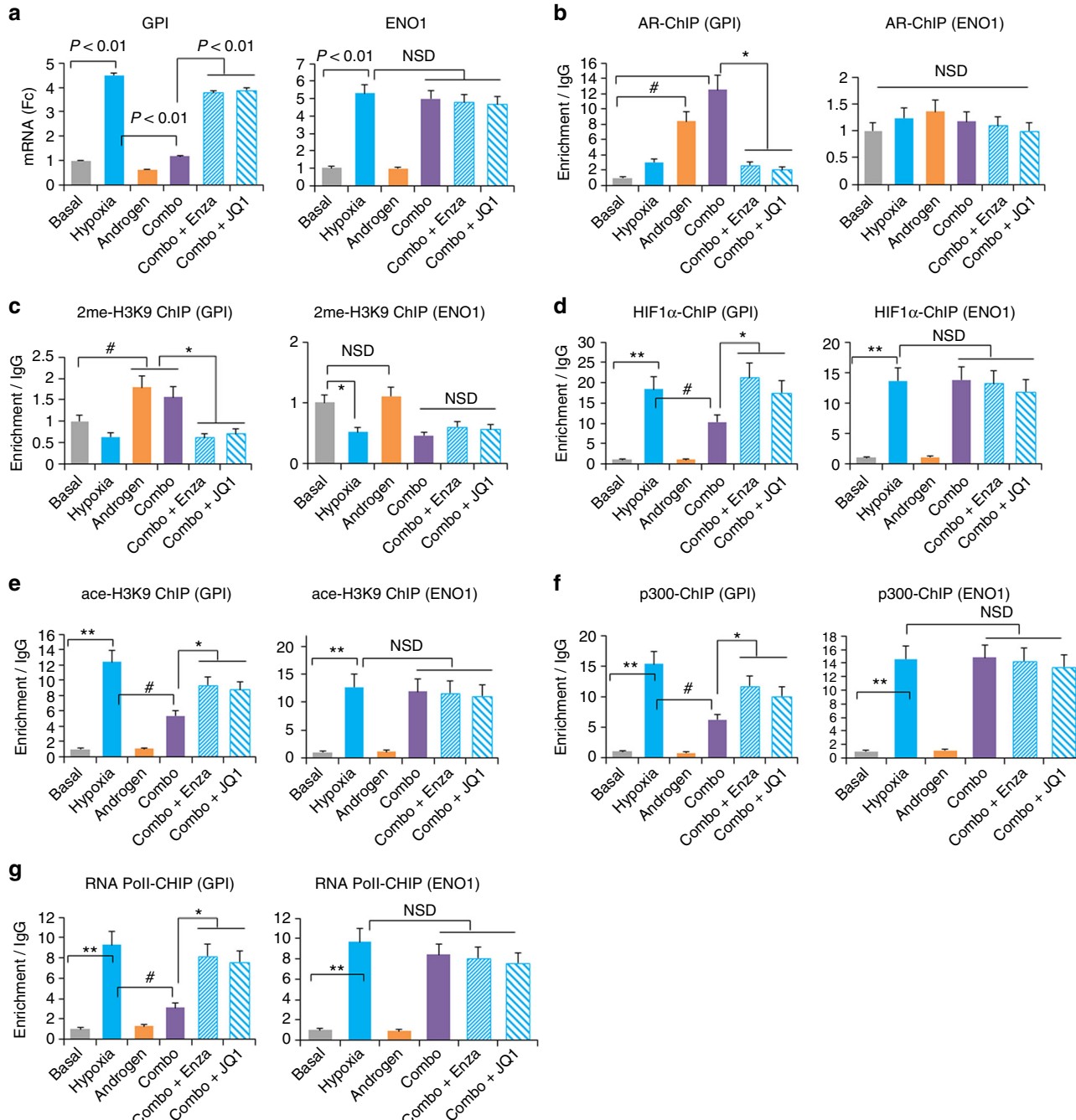

**Fig. 4** Androgen/AR epigenetically attenuates HIF1-mediated GPI transcription in hypoxia. LNCaP-AdtHs cells were treated with basal (CSS media), 1% O2 (hypoxia), 1 nM R1881 (androgen), hypoxia + androgen (Combo), or Combo + androgen/AR-targeted agents (enzalutamide or JQ1) for 48 h. Afterwards, gene expressions (GPI and ENO1) were determined by qRT-PCR and the enrichment of epigenetic compositions at the HRE of GPI and ENO1 promoters was determined by ChIP. **a** The mRNA level of GPI and ENO1 at the indicated conditions. **b** Levels of AR enrichment at the HRE regions in GPI and ENO1 promoters. **c** Levels of 2-methyl histone 3 lysine 9 enrichment at the HRE regions in GPI and ENO1 promoters. **d** Levels of HIF1α binding to the HRE regions in GPI and ENO1 promoters. **e** Levels of acetylation of histone 3 lysine 9 enrichment at the HRE regions in GPI and ENO1 promoters. **f** Levels of p300 enrichment at the HRE regions in GPI and ENO1 promoters. **g** Levels of RNA polymerase II enrichment at the HRE regions in GPI and ENO1 promoters. All values were mean ± s.d. relative to the basal condition, $n = 3$, #, *$P < 0.05$, **$P < 0.01$, NSD, two-sided $t$-test

model, tumor cells remain androgen/AR-dependent in normoxia, and can be effectively inhibited by the targeted treatment. However, subsets of tumor cells develop resistance in hypoxia because they are more capable of adaptively responding to androgen/AR-blockade in the hypoxic condition, becoming androgen/AR-independent, or conditionally (hypoxic) androgen/AR-independent.

Thousands of metastatic cancer patients are undergoing multiple forms of androgen/AR-blockade treatments, e.g., standard ADT/castration, or the next-generation drug enzalutamide. Intratumor hypoxia occurs frequently. Therefore, it is pathologically possible that a subset of cancer cells is capable of conditional (hypoxic) androgen/AR-independence, and is selected in the resistant tumor by all types of androgen/AR-blockade

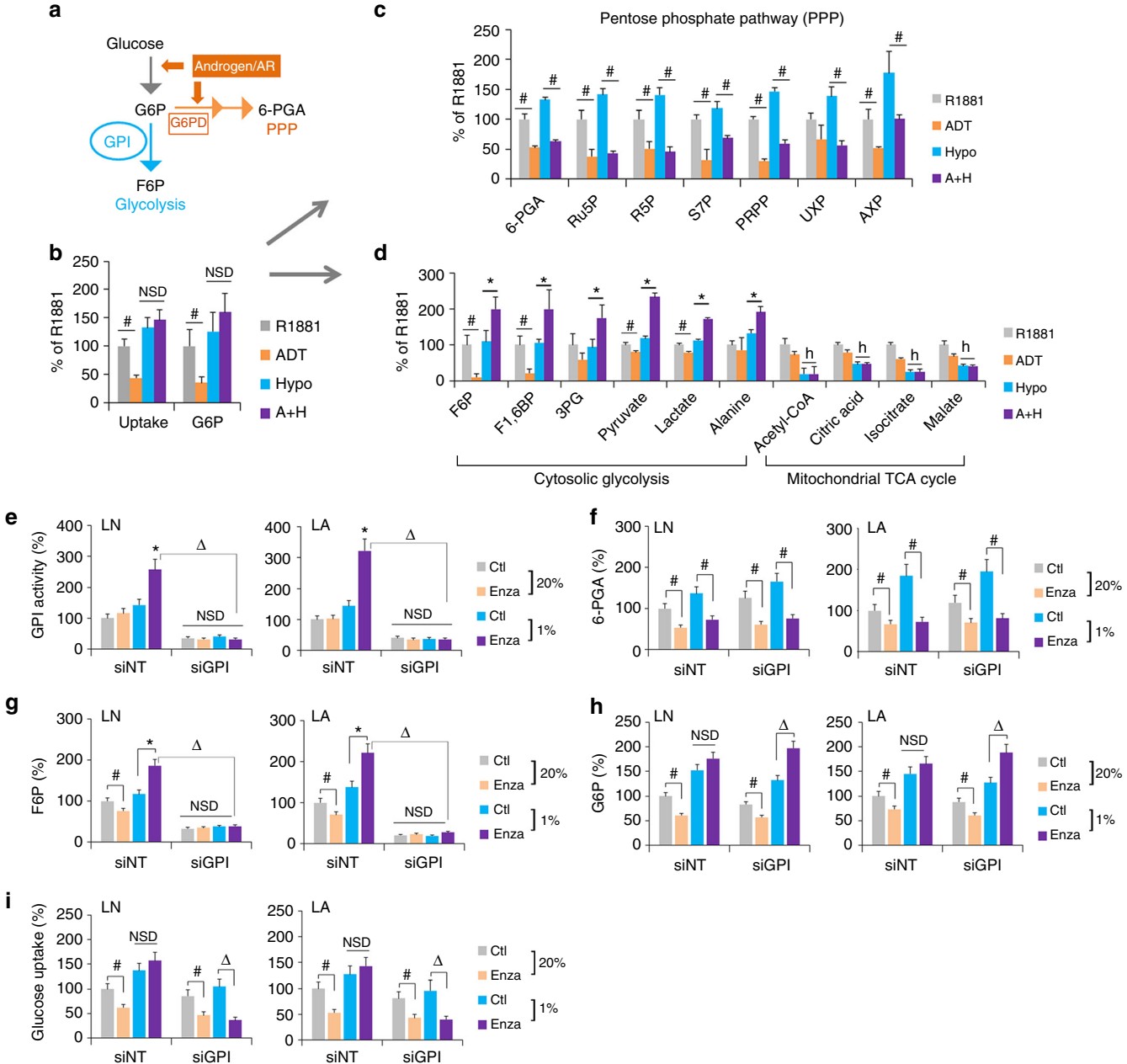

**Fig. 5** ADT or enzalutamide in hypoxia rewires glucose metabolism from PPP to glycolysis through GPI. **a** Schematic representation of glucose metabolism that can be directed to PPP by G6PD or to glycolysis by GPI. Androgen/AR is known to have positive effects on PPP, including the increase of glucose uptake, conversion to G6P, and flux to PPP via G6PD[39,40]. **b–d** LNCaP-AdtHs cells were cultured in conditions of R1881 (1 nM), ADT (5% CSS), hypoxia (1% $O_2$), or ADT + hypoxia (A+H). The glucose metabolic flux was evaluated for glucose uptake and phosphorylation to G6P (**b**), for pentose phosphate pathway (**c**), and for glycolysis pathway (cytosolic glycolysis) and the mitochondrial TCA cycle (**d**). The glucose uptake was determined with a commercial kit (BioVision). The metabolic flux was determined by incubating cells with [U-$^{13}$C]-labelled glucose, and measuring levels of $^{13}$C-glucose metabolites with electrophoresis-mass spec (EC-MS) or $^1$H NMR. The level of glucose metabolites was adjusted by the viable cell number and expressed as % to the R1881 condition, mean ± s.d., $n = 3$, *,#$P < 0.05$ ADT + hypoxia vs. R1881 + hypoxia, #$P < 0.05$ ADT vs R1881, h$P < 0.05$ hypoxia vs non-hypoxia, two-sided $t$-test. **e–i** The effect of GPI-siRNA on glucose metabolic pathways based on PPP-metabolite 6-PGA vs. glycolysis metabolite F6P, in AdtHs cells from LNCaP (LN) or LAPC4 (LA). Cells were cultured in media with CSS + 1 nM R1881, transfected with siRNA (siNT or siGPI), and then treated by solvent control (Ctl) or 10 μM enzalutamide in normoxia (20%) or hypoxia (1%) for 48 h. The enzymatic activity of GPI (**e**), PPP-metabolite 6-PGA (**f**), glycolysis/GPI-metabolite F6P (**g**), the upstream metabolite and substrate for both G6P (**h**), and glucose uptake (**i**) were all determined by commercially available kits from BioVision. All values were adjusted with viable cell numbers and expressed as mean + s.d. relative to siNT with solvent control in 20% $O_2$, $n = 3$, *$P < 0.05$ enza + hypoxia vs. Ctl + hypoxia, #,Δ$P < 0.05$, NSD, two-sided $t$-test

strategies (Supplementary Fig. 12). Because most genomic analyses of clinical tumors do not differentiate cells based on oxygen concentration, signals from hypoxic cells can be diluted or masked by non-hypoxic cells. Therefore, the clinical validation of our model requires new approaches focusing on the hypoxic subset and the intratumor heterogeneity.

The molecular basis of our conditional (hypoxic) androgen/AR-independence model is based on our observation that a subset

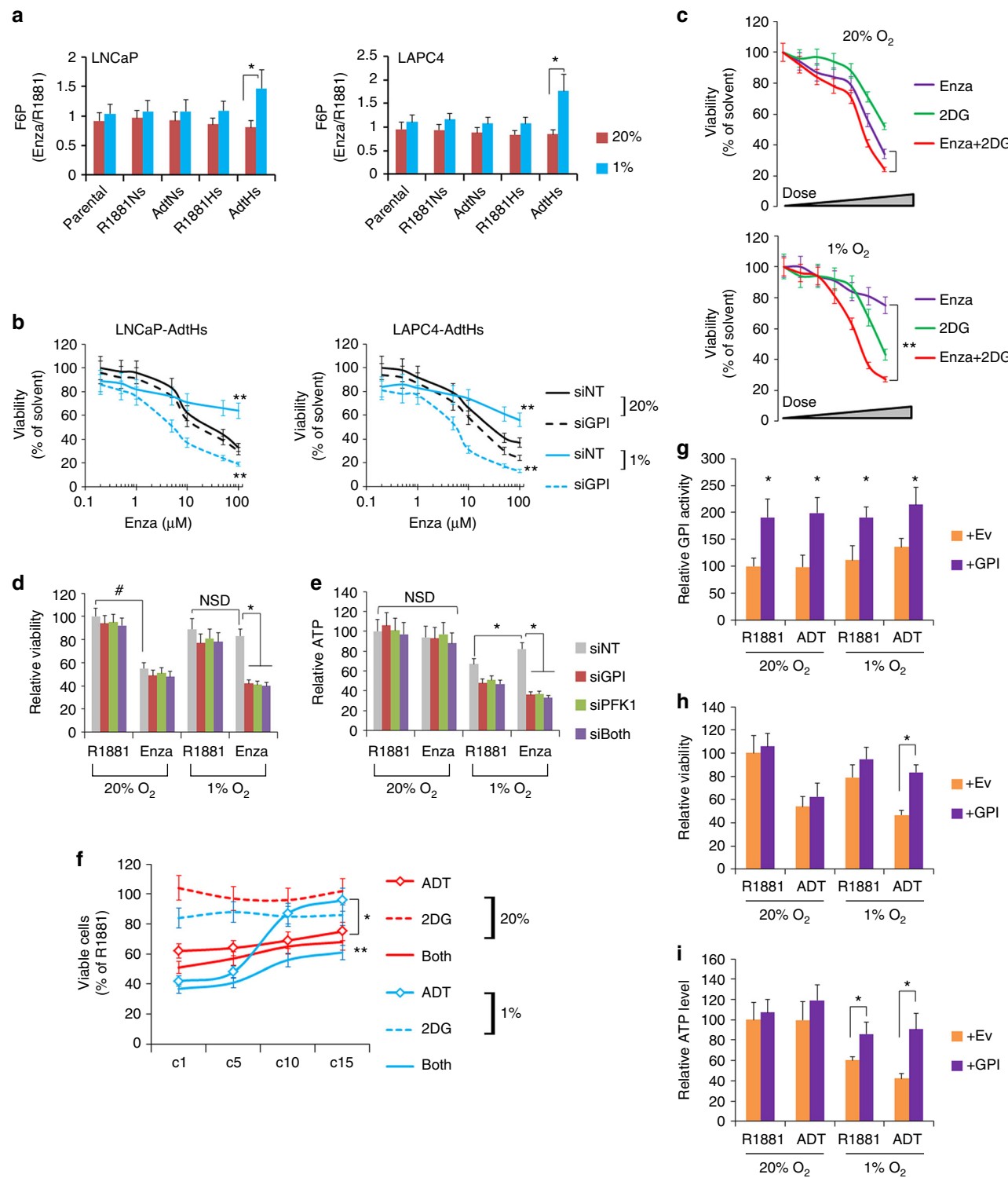

of hypoxia-response genes, e.g., GPI, critical to cellular adaptation and survival to hypoxia stress are attenuated by androgen/AR, but restored by androgen/AR-targeted therapy in hypoxia. AR in prostate cancer is capable of exerting overwhelmingly and multifaceted oncogenic activities, including the promotion of glucose metabolism[39,40,43]. Counterintuitively, it also negatively regulates some oncogenic pathways, e.g., loss of PTEN and gain of PI3K/AKT[44,45]. Our finding suggests that this negative regulation extends to hypoxia/HIF1. It is possible that the hypoxia/HIF pathways, albeit oncogenic, are secondary to AR in driving

prostate cancer. When the androgen/AR-axis is inhibited by the targeted therapy, however, the negative regulation provides the molecular basis for adaptive escape and survival response.

Based on this study and works of others[39,40,43], we conclude that androgen/AR promotes glucose metabolism via both the biosynthetic pentose pathway and the glycolytic pathway in normoxia. It does so by promoting the expression of multiple genes/enzymes along the glycolysis and pentose pathways. Androgen/AR-targeted inhibition in normoxia blocks the glucose metabolism, reducing both PPP (6-PGA) and glycolytic (F6P)

**Fig. 6** GPI reduces the antitumor activity of enzalutamide in hypoxia in vitro. **a** Relative levels of F6P in cells treated by enzalutamide (Enza) vs solvent control (R1881) in normoxia or hypoxia. Cells were cultured in CSS + 1 nM R1881, and then treated with solvent control or 10 μM enzalutamide in 20% or 1% $O_2$ for 48 h. F6P was measured and expressed as mean ± s.d. relative to R1881 (Enza/R1881), $n = 3$, *$P < 0.05$, two-sided $t$-test. **b** Dose-dependent inhibition of viability/growth by enzalutamide against AdtHs cells in 20% or 1% $O_2$ with/without GPI-siRNA. LNCaP and LAPC4-AdtHs cells, cultured in CSS media + 1 nM R1881, were transfected with siRNA (siNT or siGPI), and treated with increasing doses of enzalutamide for 96 h in normoxia or hypoxia. Viable cells were determined by SYTO 60. The siNT vs. siGPI in hypoxia are represented by solid vs. dotted blue lines, respectively. All values are mean ± s.d. relative to 0 μM enzalutamide, $n = 3$, **$P < 0.01$, two-sided $t$-test. **c** Growth/viability of LNCaP-AdtHs cells in normoxia or hypoxia, being treated by increasing doses of enzalutamide, 2DG (0 – 10 mM), or the combination for 96 h. Viable cells were determined by SYTO 60. All values are mean ± s.d. relative to solvent control, $n = 3$, **$P < 0.01$, two-sided $t$-test. **d**, **e** LNCaP-AdtHs cells, cultured in CSS media + 1 nM R1881, were transfected with siRNA (siNT, siGPI, siPFK, or siGPI + siPFK), and treated with solvent (R1881) or enzalutamide in normoxia or hypoxia for 96 h. Viable cells were determined by SYTO 60 (**d**). ATP levels were determined by colorimetric assays (**e**). All values were mean ± s.d. relative to the siNT-R1881-20% $O_2$ condition, $n = 3$, *,#$P < 0.05$, NSD, two-sided $t$-test and/or repeated measures ANOVA. **f** Growth/viability of LAPC4 cells during the chronic-treatment with ADT, 2DG or ADT +2DG in normoxia or hypoxia. Throughout the treatment cycle ($c1 – c15$), LAPC4 cells, cultured in CSS + R1881 media, were treated by negative control (R1881), ADT, 2DG (1 mM), or ADT+2DG in normoxia or hypoxia. Cell growth after cycle 1, 5, 10, and 15 were determined and expressed as mean ± s.d. as % relative to the R1881-normoxia condition, $n = 3$, *ADT in hypoxia (blue -o- line) vs ADT in normoxia (red -o- line), **ADT in hypoxia (blue -o- line) vs ADT + 2DG (blue solid line) in hypoxia, $P < 0.05$ two-sided $t$-test and/or repeated measures ANOVA. **g**–**i** LAPC4 cells were transfected with Ev or GPI overexpression plasmids and then cultured with 1 nM R1881 or ADT in normoxia or hypoxia for 72 h. The GPI activity was determined with a kit from BioVision (**g**), growth/viability was determined with SYTO 60 (**h**), ATP level was determined with a kit from BioVision (**i**). All values are mean ± s.d. relative to the Ev-R1881-20% $O_2$ condition, $n = 3$, *$P < 0.05$, two-sided $t$-test

metabolites, which in part explains the growth inhibitory effect. In hypoxia and without the androgen/AR blockade, the PPP is maintained and even increased; androgen/AR transcriptionally attenuates the hypoxia-induced GPI gene upregulation, thereby restraining the glycolytic pathway. Importantly, androgen/AR-blockade (ADT, enzalutamide or siAR) in hypoxia blocks PPP, but restores the hypoxic upregulation of GPI, which rewires the blocked-PPP flux to the GPI-mediated glycolysis, thereby maintaining glucose metabolism

We further conclude that this metabolic switch (or metabolic plasticity), induced by androgen/AR blockade + hypoxia, may in turn promote hypoxia-induced adaptation and survival, thereby conferring androgen/AR-independence and overriding the antitumor activity of androgen/AR-blockade. One possible explanation is that the glucose metabolism in hypoxia is essential to cancer cell survival and growth. This is because cellular ATP and energy homeostasis are primarily maintained by glucose metabolism via glycolysis in hypoxia. In contrast, other energy-producing pathways, e.g., glutamine and fatty acids, require mitochondrial oxidative phosphorylation, which is largely inactive in hypoxia due to the lack of oxygen as the essential substrate[36,46]. Therefore, in response to androgen/AR-targeted therapy in hypoxia, prostate cancer cells most capable of redirecting glucose flux from PPP to glycolysis may maintain glucose metabolism and utilization, thereby gaining survival and growth advantage and evading therapy. This in part explains that overexpression of GPI confers ADT resistance in hypoxia, but not in normoxia. GPI also has glucose metabolism independent activities (moonlighting functions) by acting as an extracellular protein promoting cell growth and motility[47,48]. Thus, the GPI-mediated treatment resistance may also be mediated by an autocrine and paracrine mechanism independent of glycolysis. However, our current in vitro and in vivo data with glycolytic inhibitor 2DG strongly suggests that the GPI-mediated glycolysis pathway has a central role in conferring resistance in the AdtHs cells. Hypoxia response is heterogenic[49–51], thus this adaptive response to therapy in hypoxia can also be heterogenic—only subsets of tumor cells more capable of expressing GPI are more likely to be enriched or selected as therapy-resistant clones. This may explain that level of GPI increase by enzalutamide + hypoxia was the highest in the resistant AdtHs cells derived from chronic-ADT-in hypoxia.

For the first time, this study establishes the negative interplay/crosstalk between androgen/AR and hypoxia/HIF1. It is well

known that hypoxia upregulates glucose metabolism via increasing glycolysis in many types of normal and cancer cells. Based on our observation in AR+ prostate cancer cells, however, we conclude that glycolysis is not significantly upregulated by hypoxia due to the negative interplay between AR and HIF1 at the promoter of GPI. We found that the androgen/AR-response/binding element, AREhs, was adjacent to the hypoxia/HIF-response/binding element, HRE, in the promoter region of GPI. Our data from ChIP and reporter assays suggest the possibility, which AR first binds to the AREhs in normoxia, and then directly binds to the adjacent HRE in hypoxia, possibly through the already documented AR-HIF1 protein-protein interaction[52,53]. The AR occupancy at HRE may reduce HIF1 binding and HIF1-mediated local chromatin modifications in favor of transcriptional activation, hence the attenuated-transcriptional upregulation. It may also suggest that the expression of GPI is independent of and negatively associated with the androgen/AR-axis. Interestingly, the clinical prevalence of GPI overexpression is significantly higher in patients with androgen/AR-negative neuroendocrine prostate cancer. Also, the clinical observation of AR binding to HRE was made with AR-based ChIP-sequencing in CRPC samples[54].

We also observed that hypoxia and androgen can have positive crosstalk, in which these two factors work collaboratively to regulate the gene expression and cancer hallmark pathways in additive or synergistic fashions. This is in agreement with the current established-concept that androgen signaling is capable of enhancing hypoxia gene expression and vice versa[32,33]. Although this study focuses on the negative crosstalk, the positive crosstalk may provide the molecular mechanism by which hypoxia facilitates androgen/AR-signaling to drive prostate cancer development both in the primary site and in the metastatic bone environment. Further, male hormones and hypoxia both have important roles in other forms of human cancer and cardiovascular diseases. Thus, the positive crosstalk between hypoxia and androgen may lead to new molecular understanding and treatment.

## Methods

**Cell lines**. Cell lines used in this study are human prostate cancer cell lines, LAPC4, LNCaP, and DU145. All were purchased from American Type Culture Collection (ATCC). The DU145 cell line with stable AR transfection is a gift from Dr. John Isaacs at The Johns Hopkins University. All cell lines are routinely checked for mycoplasma contamination with the detection kit from Thermo Fisher.

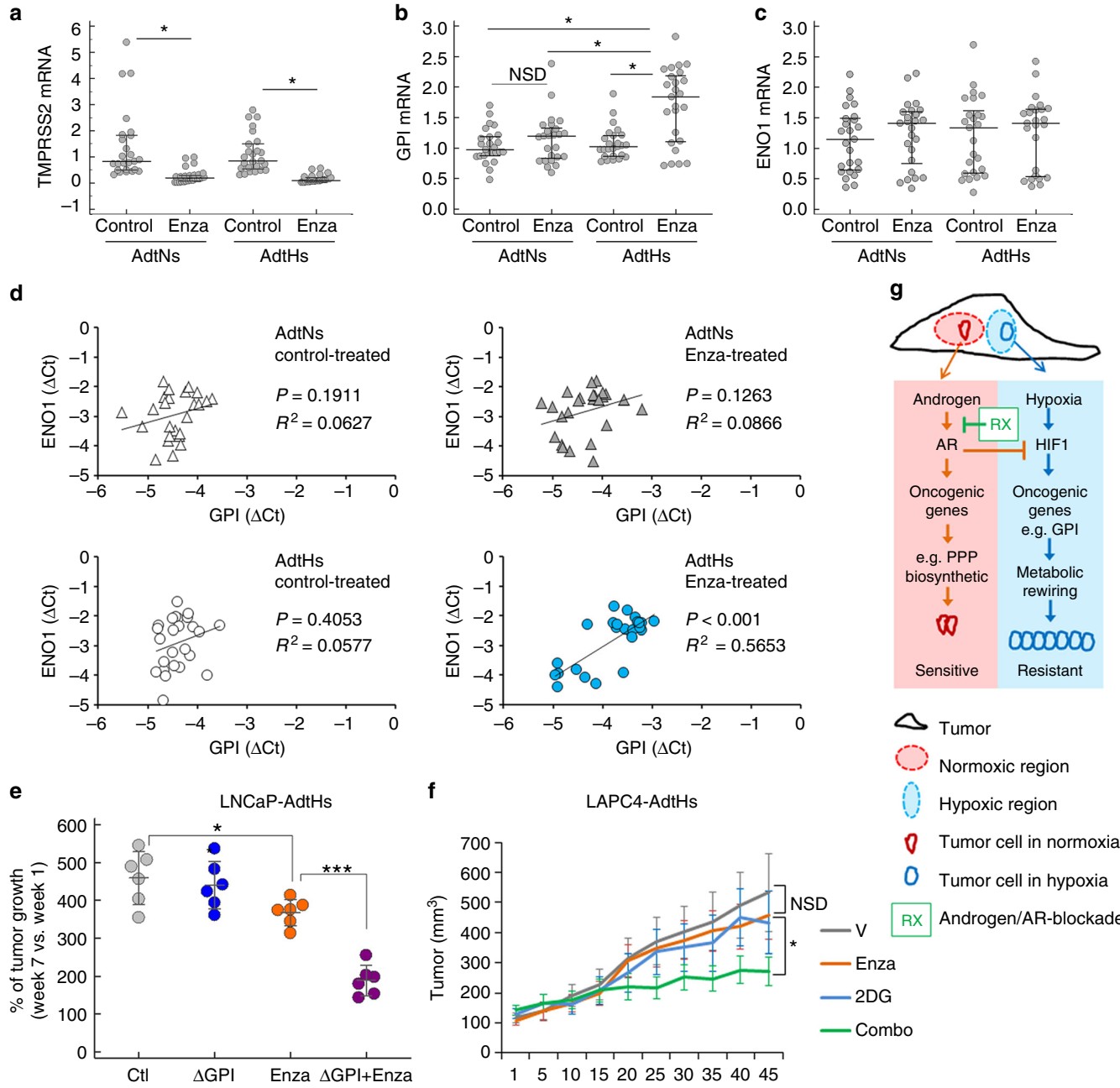

**Fig. 7** GPI reduces the antitumor activity of enzalutamide in vivo. **a–c** Xenograft tumors from LNCaP AdtNs or AdtHs cells in Fig. 1g were cut into small pieces and used for RNA isolation, cDNA synthesis, and qRT-PCR. On average, 4 different locations (4 pieces) of each tumor were sampled, the sample size for each group (solvent or enzalutamide-treated AdtNs or AdtHs) is 25 ($n = 25$). The mRNA expression of TMPRSS2 (**a**), GPI (**b**), and ENO1 (**c**) in each tumor piece (dot) is relative to the average of AdtNs tumors from solvent-treated control mice. Median (center line) ± 25 percentile, $n = 25$, *$P < 0.05$, two-sided $t$-test in **a** or repeated measure ANOVA in **b**. **d** The correlation between ENO1 and GPI mRNA expression in each tumor piece, grouped by AdtNs vs. AdtHs and by control-treated vs. enzalutamide. The $P$-value and $R^2$ were calculated by linear regression, $n = 25$. **e** Antitumor effect of enzalutamide in vivo with/without GPI. LNCaP-AdtHs cells with CRISPR/Cas9 knockout of GPI (ΔGPI) or negative control were injected subcutaneously to male nude mice. After the tumors were palpable, mice were treated with vehicle (Ctl) or enzalutamide for 7 weeks. Tumor volume was followed by digital caliper. The % of growth for each tumor (post-treatment/pre-treatment) is shown as mean (center line) ± s.d. $n = 6$, *$P < 0.05$, ***$P < 0.01$, two-sided $t$-test. **f** Antitumor effect of enzalutamide in vivo in the presence/absence of glycolysis inhibitor 2DG. LAPC4-AdtHs cells were injected subcutaneously to male nude mice. After the tumors were palpable (~100 mm³), mice were treated with vehicle (V), enzalutamide, 2DG, or enzalutamide + 2DG for 6 weeks. The tumor volume is mean ± s.d. $n = 6$, *$P < 0.05$, NSD, two-sided $t$-test and/or repeated measures ANOVA. **g** Schematic diagram of the negative regulation of hypoxia/HIF1 gene expression, e.g., GPI, by androgen/AR, leading to hypoxia-induced adaptive resistance to androgen/AR-targeted therapy (conditional androgen/AR-independence and therapy resistance)

**Key reagents and cell culture conditions**. Androgen-dependent prostate cancer cell lines LNCaP and LAPC4 were purchased from ATCC. Enzalutamide (MDV3100) and JQ1 were purchased from Sigma-Aldrich.

Cells were cultured in RPMI or DMEM media with 5% charcoal-stripped FBS (CSS) and 1% penicillin streptomycin. Synthetic androgen R1881 was added at final concentration of 1 nM as androgen supplements. Four conditions were routinely used: (1) Basal (castration or ADT) condition, media with 5% charcoal-striped serum (CSS), (2) Hypoxia, 1% oxygen, (3) Androgen, CSS + 1 nM R1881, and (4) Combination (combo), hypoxia + androgen. In the anti-androgen therapeutic experiments, cells were cultured in the androgen condition (CSS + 1 nM R1881), and were treated by anti-androgen/AR drug enzalutamide (Enza) at the final concentration of 10 μM. The hypoxia or 1% oxygen condition was created in the cell culture incubator by replacing oxygen with compressed nitrogen. Sodium Bicarbonate (30 mM) was used to neutralize the hypoxia-induced lactate acid for experiments without involving metabolic measurement.

**Chronic-treatment-in-hypoxia or -normoxia (CTIH or CTIN)**. For chronic-ADT treatment/selection, LNCaP or LAPC4 cells were treated with 15 rounds of androgen deprivation therapy (ADT) in normoxia (20% $O_2$) or hypoxia (1% $O_2$). For each round, cells were plated in RPMI or DMEM media with 5% CSS +1 nM synthetic androgen R1881. When the cells reached ~60% confluency, fresh media without R1881 was added to the cells to start ADT, which lasted 48 h in normoxia or hypoxia. After each round, the surviving cells were passaged and re-plated to receive the same treatment in the next round. After 15 rounds, cells were labeled based on the treatment condition.

**Microarray, differential expression, clustering, and informatics**. Gene expression profiles were determined at the OHSU Gene Profiling Shared Resource, with Affymetrix PrimeView Human Gene Expression Array, consisting of ~48,889 gene probes and representing ~20,014 genes. Total RNA was collected from cells cultured in four conditions (basal, hypoxia, androgen, and hypoxia + androgen), and each had a biological triplicate ($n = 3$). False-discovery (FDR) adjusted $t$-test was used to determine the differential expression of individual gene probes between any two conditions. The level of significance (differentially expressed) was set at $\Delta\log_2$ mRNA > 0.75 or <−0.75 with the FDR-adjust $P$-value ($q$) < 0.05. The log2 expression of all gene probes with $q$ < 0.05 was mean-centered and clustered by complete linkage hierarchical clustering with Cluster 3.0, which was then visualized as heatmap with Jave TreeView. The microarray data was also uploaded into the GSEA software, and the enrichment of cancer hallmark pathways was determined with FDR-$q$ < 0.25.

**Gene knockdown, knockout, and overexpression**. siRNA was carried out to transiently silence HIF1α, HIF2α, AR, and GPI. To effectively target each gene, siRNA cocktails containing up to 4 constructs against each gene were purchased from Sigma. Briefly, cells were transfected with silencing siRNA or non-targeting control siRNA for 24 h, and then moved to normoxic or hypoxic conditions to start the experiment. The stable knockout of GPI was achieved with the CRISPR/Cas9 system from Thermo Fisher Scientific. The GPI overexpression plasmid was purchased from OriGene. Efficacies of siRNA, CRISPR/cas9 and overexpression of GPI were all determined by western blots and GPI enzymatic activity assay.

**Western blotting, qRT-PCR, and ChIP-PCR**. The antibodies for western blots and ChIP were purchased from Abcam (HIF1α, GPI), Santa Cruz (AR, p300, tubulin), Novus Biological (HIF2α, HIF1β), and Active Motif (2me-H3K9, ace-H3K9, RNA polymerase II). The western blot was carried out with equal amount of whole cell lysates and visualized with LiCor Odyssey Fluorescence Imager. Tubulin was used as the equal loading control. The unprocessed blots are included in Supplementary Fig. 13.

The qRT-PCR was carried out with TRIzol-extracted RNA and SYBR-Green-based detection. The expression of β-actin was used as the control. All RT-PCR primers were purchased from Real Time Primers LLC, and have been verified for human RT-PCR. All primer sequences are included in Supplementary Fig. 14.

ChIP assay was carried out with a kit from Active Motif according to the manufacture protocol, in which chromatins from at least $10^6$ cells were isolated, and immunoprecipitated with antibodies against HIF1α, AR, 2me-H3K9, ace-H3K9, p300, RNA polymerase II, and rabbit or mouse IgG as the negative control. The immunoprecipitated DNA was then analyzed by SYBR-Green-based qRT-PCR for the promoter HRE of GPI and ENO1. The primer sequences for the GPI promoter are forward 5′-CTTGGTTTTCCTCAATAGCCCT-3′, reverse 5′-CCT GTGCACTAGTCGGCTTC-3′; for ENO1 promoter are 5′-AGTTGTCAGCAAG GTCGAGG-3′, reverse 5′-AACTCTGGCCCCAGATAGGA-3′. The PCR signal was adjusted with the IgG and expressed as fold of enrichment relative to the basal normoxic condition.

**Site-directed mutagenesis and promoter-reporter assays**. Plasmids encoding luciferase reporter gene under the control of the promoter of GPI was purchased from SwitchGear Genomics. PCR-based site-directed mutagenesis was carried out to mutate the AREhs sequences in the GPI promoter region. The primers to generate AREhs mutant are: forward 5′-CACAGCGCCTACATCACTGCCTGT

AAC-3′ and reverse 5′-GTTACAGGCAGTGATGTAGGCGCTGTG-3′. The wild-type or mutant reporter plasmids were co-transfected with constitutive GFP expressing plasmid into LNCaP or DU145 cells. After treatment, the luciferase activity was determined by an assay kit from SwitchGear Genomics, adjusted by GFP, and expressed as fold change relative to basal and normoxia.

**Viability/proliferation/cell cycle**. Cell viability and proliferation experiments were carried out in 96- or 24-well format. Viable cells were determined with SYTO 60 fluoresce staining (Thermo Fisher), quantitated by LiCor Odyssey Imager, and expressed as % of solvent-treated control. Cell cycle distribution was determined by propidium iodide-based flow cytometry with the sub-G0/G1 population being considered as cell death.

**Metabolic analysis**. The metabolic flux of [U-$^{13}$C]-labelled glucose (Cambridge Isotope Laboratories) in LNCaP-AdtHs cells in response to R1881 or ADT in 20% or 1% $O_2$ was determined by both Capillary Electrophoresis-Mass Spec (EC-MS) and $^1$H NMR. The EC-MS analysis was carried out by Human Metabolome Technologies (HMT)[55,56]. The NMR analysis was carried out at the Oregon State University NMR core facility[57,58]. With both analyses, levels of $^{13}$C-labelled glucose metabolites were adjusted by viable cell numbers, and expressed as % relative to R1881/20% $O_2$. Colorimetric and fluorescent assay kits from BioVision were used to determine the enzymatic activities of GPI (Catalog # K775), and the levels of 6-PGA (# K217), F6P (# K689), G6P (# K675), ATP (# K354), and glucose uptake (# K676). All procedures were based on the manufacture's protocol. The F6P signal in each sample was adjusted by the background due to potential crossover of G6P using non-enzyme control buffer. All signals were adjusted with total viable cell numbers and/or protein concentrations determined in parallel experiments.

**Xenograft experiments**. All animal experiments are in compliance with protocols approved by OHSU and Johns Hopkins IACUC. Subcutaneous implants of LNCaP and LAPC4 cells were generated in male nude mice by injecting ~5 million cells mixed with Matrigel[59]. The dose for enzalutamide was 10 mg/kg (gavage), 5 times per week. The dose for 2DG was 200 mg/kg (i.p.), three times per week. The tumor volume was determined by digital caliper measurement, and expressed either as mm$^3$ (volume = width × depth × length × 0.52) or as % of growth relative to the start of treatment.

**cBioPortal (http://www.cbioportal.org) analyses**. The expression of androgen-attenuated hypoxia-response genes, including GPI, were determined in six prostate cancer patient cohorts—Neuroendocrine Prostate Cancer (Trento/Cornell/Broad 2016, $n = 114$), Prostate Adenocarcinoma (Broad/Cornell, Nat Genet 2012, $n = 112$), Prostate Adenocarcinoma (Fred Hutchinson CRC, Nat Med 2016, $n = 176$), Prostate Adenocarcinoma (MSKCC, Cancer Cell 2010, $n = 216$), Prostate Adenocarcinoma (TCGA, $n = 499$), and Metastatic Prostate Cancer, SU2C/PCF Dream Team (Cell. 2015, $n = 150$). The Z-score threshold was set at ±2.

**Statistical analysis**. All experimental data were expressed as mean and standard deviation (s.d.) unless indicated otherwise. Statistical comparisons between two sample sets were performed with two-sided student's $t$-test or paired $t$-test, comparisons among more than two samples were performed with repeated measures ANOVA, using MedCalc software. $P < 0.05$ was considered as significant, and $P \geq 0.05$ was considered as not significant different (NSD).

## Data availability

The microarray data generated with this study have been deposited in Gene Expression Omnibus (GEO) database under the accession GSE120343. All data are available to the public. The prostate cancer TCGA dataset referenced in the study are available in a public repository from the cBioPortal website (http://www. cbioportal.org/). The authors declare that all other data supporting the findings of this study are available within the article and its supplementary information files and from the corresponding author upon reasonable request.

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

## Acknowledgements

We thank Dr. Huafeng Zhang at University of Science and Technology of China for providing helpful protocols and discussions; members from George Thomas, Joshi

Alumkal, Mu-Shui Dai labs at OHSU and Sushant Kachhap labs at JHU for advice in experimental methods and bioinformatic analysis; Dr. Dennis Koop and Xiangshu Xiao at OHSU for helpful discussions on glucose metabolites extraction and analysis; Dr. Chris Harrington and Kristine Vartanian at OHSU Gene Profiling Shared Resource for cDNA microarray analyses; and Sunil Joshi for editing. The Oregon State University NMR facility is supported in part by NIH HEI grant 1S10OD018518 and M.J. Murdock Charitable Trust grant #2014162. This work was supported by Prostate Cancer Program at OHSU and NIH Grants 5R01CA149253 and 5R01CA207377 to DZQ.

## Author contributions

D.Z.Q. designed all the experiments with contributions from H.G., T.M.B., S.K.K., and M.-S.D. H.G., C.X., Q.L., and J.M. performed most experiments with contributions from X.-X.S., Y.C., K.Q., V.H., A.C., F.P., J.Y., S.D. P.N.R. performed 1H NMR. D.Z.Q., H.G., S.K.K., J.M. and M.-S.D. performed data analysis and bioinformatic analysis. DZQ wrote the paper with contributions from H.G., J.M., T.M.B., S.K.K., and M.-S.D.

## Additional information

**Competing interests:** The authors declare no competing interests.

