## [Peer Review File · Nature Communications]

Reviewers' Comments:

Reviewer #1:

Remarks to the Author:

The study entitled "The interplay between hypoxia and androgen at glucose-6-phosphate isomerase controls a metabolic switch to reduce the antitumor efficacy of androgen/AR-targeted therapy" addresses two areas in prostate cancer biology, androgen receptor (AR) signaling and hypoxia, that are of great interest to the field but that are rarely studied together. Here, the authors demonstrate that androgen-deprivation therapy (ADT) combined with hypoxia promote the regulation of a subset of genes that contribute towards resistance to the 2nd generation AR antagonist, enzalutamide. Interestingly, AR represses the hypoxia-mediated expression of phosphoglucose isomerase (GPI), a glycolytic enzyme. Hence, under conditions of hypoxia, enzalutamide increases GPI expression, promoting pro-cancer glycolysis as a resistance mechanism. Overall, this is a very well written manuscript that addresses topics that will be of great interest to the broad fields of prostate cancer and cancer metabolism. Importantly, it describes a potential mechanism of drug resistance and therefore sets the stage for novel approaches that could be tried to overcome resistance to AR blockade. The study is supported by exceptional mechanistic studies elucidating how AR and hypoxia modulate GPI expression and activity, xenograft mouse models and clinical correlation data. The main weakness is that there are currently no analyses of metabolic flux. This is important because one of the key points is that this alteration in GPI levels is hypothesized to "reroute" metabolism. Given the number of routes glucose metabolism can take, it is unclear if the changes observed are due to increases in influx or decreases in efflux. Further, there were some missed opportunities where the in vivo tumor samples could have been interrogated more to strengthen the authors conclusions. Below is a more detailed list of issues that should be addressed to further strength this promising study prior to publication.

Major points:

- 1) The data supporting the model of metabolic rerouting should ideally be proven using metabolic flux/tracing approaches. This could be done in the cell models and would prove that glucose is being diverted away from the pentose phosphate pathway (PPP) towards glycolysis. In addition, it could be used to determine whether the carbons continue on into the TCA cycle or are converted to lactate (or possibly other pathways including the hexosamine biosynthetic pathway or non-oxidative arm of the PPP, etc). At present, the data is largely looking at snap shots of metabolites that could be increased from either increased production or decreased breakdown, making it difficult to draw conclusions.
- 2) Related to the point above, contrary to what is stated by the authors, GPI is not widely considered the committed step of glycolysis. GPI catalyzes a reversible reaction. Hence, simply demonstrating increased GPI activity does not necessarily mean more forward flux through glycolysis. Further, the product of GPI, fructose-6-phosphate (F6P), can also enter (or come from) the non-oxidative arm of the PPP. These are additional reasons to do the tracing studies. However, it is noted that the experiments demonstrating knockdown of PFK1, which is a canonical, committed step of glycolysis, do support the authors model.
- 3) In addition to its effects in glycolysis, GPI has known moonlighting functions outside of the cell that have been demonstrated to contribute towards cancer metastases. This alternative mechanism of action was not considered and should, at a minimum, be addressed in the Discussion.
- 4) It is surprising that the transient transfection of LAPC4 cells, which typically have a low transfection efficiency, with a GPI expression plasmid was sufficient to promote a significant level of therapy resistance in a population of cells. Controls should be included to demonstrate transfection efficiency and confirm GPI overexpression.
- 5) GPI levels or activity should be assessed in the tumor samples from Figure 1G to support the model.
- 6) Given that the functional effects of GPI is a major point of this study and concerns regarding the

off-target effects of chemical siRNAs, the experiments described in Figures 5D-G and 6B should be confirmed using multiple siRNAs, an add-back approach or the CRISPR GPI knockout line created and used in Figure 6G.

7) While potentially beyond the scope of this study, evidence of increased glycolysis in the resistant tumors from any of the animal studies would greatly strengthen the data.

Minor points:

- 1) JQ1 is not a direct AR inhibitor. Rather it indirectly targets AR via associated BRD4.
- 2) It is not clear why repeated cycles of ADT were needed compared to continuous ADT.
- 3) Comment: It is worth noting that subcutaneous injection sites are quite hypoxic and hence, without stating so, the authors chose an ideal xenograft model site for their studies. While certainly beyond the scope of this study, it would be interesting to see if similar effects would be observed at other, less hypoxic sites.
- 4) It is premature to conclude that ATP production and energy homeostasis are the main reasons why glycolysis would be driving disease progression in this context. This was never formally tested and there are many other important aspects of glycolysis that contribute towards tumor growth and cancer spreading.
- 5) Catalog numbers for the metabolic kits should be provided because it is not clear how they assessed F6P levels. The stated company does not appear to offer an assay kit to quantitate this metabolite, which is often difficult to distinguish from G6P.
- 6) Where repeated comparisons are made, repeated t tests are likely inadequate. Rather, ANOVAs with appropriate post hoc tests should be used.
- 7) Some of the stat comparison bars are misaligned in Figure 4.
- 8) Was GPI included in any of the highlighted GSEA pathways?
- 9) Typo in Supp Fig 13B y axis.

Reviewer #2:

Remarks to the Author:

The authors have taken a novel approach to identify factors which can contribute to ADT-resistance under conditions of hypoxia. This has evolved the adaptive selection of cell-lines through repeated exposure to hypoxia and ADT. Of the prognostic genes identified in this study the authors have focussed on GPI and its contribution to sustaining glycolysis under conditions of hypoxia once androgen-mediated repression has been removed. The study is provocative and carefully undertaken with respect to GPI. The manuscript will need some minor revisions prior to publication including the full deposition of microarray data in a suitable repository. The authors have used the TCGA dataset to associate the prognostic value of genes that are induced by hypoxia and repressed by androgen but highly expressed under conditions of ADT and hypoxia. They have done so through single-gene analysis. Are all of these prognostic genes differentially expressed in the same subset of TCGA cases? Which pathways are dysregulated in cases identified by these genes? How does the GSEA analysis undertaken in supplementary figure 3 and enrichments shown here map back to these genes? Which genes account for the enrichments attributed to biologies in this analysis - eg. Androgen response or estrogen response. These should be provided in tabular form to provide extra insights. Finally, of the genes that have been reported to be prognostic in the TCGA data, including GPI, are any also known to be regulated by other poor prognosis mutant drivers of prostate cancer - eg. TP53, PTEN etc. This extra information will significantly enhance the impact of this study.

Reviewers' comments:

Reviewer #1:

*The study entitled “The interplay between hypoxia and androgen at glucose-6-phosphate isomerase controls a metabolic switch to reduce the antitumor efficacy of androgen/AR-targeted therapy” addresses two areas in prostate cancer biology, androgen receptor (AR) signaling and hypoxia, that are of great interest to the field but that are rarely studied together. Here, the authors demonstrate that androgen-deprivation therapy (ADT) combined with hypoxia promote the regulation of a subset of genes that contribute towards resistance to the 2nd generation AR antagonist, enzalutamide. Interestingly, AR represses the hypoxia-mediated expression of phosphoglucose isomerase (GPI), a glycolytic enzyme. Hence, under conditions of hypoxia, enzalutamide increases GPI expression, promoting pro-cancer glycolysis as a resistance mechanism. Overall, this is a very well written manuscript that addresses topics that will be of great interest to the broad fields of prostate cancer and cancer metabolism. Importantly, it describes a potential mechanism of drug resistance and therefore sets the stage for novel approaches that could be tried to overcome resistance to AR blockade. The study is supported by exceptional mechanistic studies elucidating how AR and hypoxia modulate GPI expression and activity, xenograft mouse models and clinical correlation data. **The main weakness is that there are currently no analyses of metabolic flux.** This is important because one of the key points is that this alteration in GPI levels is hypothesized to “reroute” metabolism. Given the number of routes glucose metabolism can take, it is unclear if the changes observed are due to increases in influx or decreases in efflux. **Further, there were some missed opportunities where the in vivo tumor samples could have been interrogated more to strengthen the author’s conclusions.** Below is a more detailed list of issues that should be addressed to further strength this promising study prior to publication.*

Response - As shown in response 1 and 2 above, these two concerns are addressed in our revision, page 8-9/Figure 5, and page 11/Figure 7, respectively.

Major points:

1) *The data supporting the model of metabolic rerouting should ideally be proven using metabolic flux/tracing approaches. This could be done in the cell models and would prove that glucose is being diverted away from the pentose phosphate pathway (PPP) towards glycolysis. In addition, it could be used to determine whether the carbons continue on into the TCA cycle or are converted to lactate (or possibly other pathways including the hexosamine biosynthetic pathway or non-oxidative arm of the PPP, etc). At present, the data is largely looking at snap shots of metabolites that could be increased from either increased production or decreased breakdown, making it difficult to draw conclusions.*

Response – as suggested by the reviewer, we performed metabolic flux/tracing of [U-13C]-glucose with EC-MS and ¹H NMR in one of the resistant prostate cancer cell clones, LNCaP-AdtHs. The result, shown in figure 5B-5D, confirms our original hypothesis that there is a glucose metabolic reprogramming from the PPP pathway, including the non-oxidative arm of PPP as seen by metabolite S7P, to the lactate-producing cytosolic glycolytic pathway in response to androgen/AR-targeted therapies in hypoxia. The glucose entry into the TCA cycle is blocked by hypoxia as seen in figure 5D. This is consistent with the well-established concept that hypoxia induce enzymes such as PDK1 to prevent glucose TCA cycle entry, while increasing the production of lactate. The glucose flow to the hexosamine biosynthetic pathway is an interesting possibility; however, we did not detect metabolites along this pathway.

2) *Related to the point above, contrary to what is stated by the authors, GPI is not widely considered the committed step of glycolysis. GPI catalyzes a reversible reaction. Hence, simply demonstrating increased GPI activity does not necessarily mean more forward flux through glycolysis. Further, the product of GPI, fructose-6-phosphate (F6P), can also enter (or come from) the non-oxidative arm of the PPP. These are additional reasons to do the tracing studies. However, it is noted that the experiments demonstrating knockdown of PFK1, which is a canonical, committed step of glycolysis, do support the author's model.*

Response – we have removed the description of GPI as the committed step. In our glucose flux/tracing analyses, we found that therapy-in-hypoxia consistently reduced all metabolites along the oxidative and non-oxidative PPP pathway, leading to a reduction of nucleotide biosynthesis (Fig 5C); therefore it is unlikely that the increase of F6P was due to the increase of non-oxidative PPP. On the other hand, the metabolites along the cytosolic glycolysis pathway, leading to the lactate production, were increased, including F1, 6BP, which is a product of PFK1. Again, this supports our hypothesis that therapy-in-hypoxia rewires the glucose metabolism from PPP to glycolysis via increasing GPI, albeit GPI is not a canonical glycolysis-committing step.

3) *In addition to its effects in glycolysis, GPI has known moonlighting functions outside of the cell that have been demonstrated to contribute towards cancer metastases. This alternative mechanism of action was not considered and should, at a minimum, be addressed in the Discussion.*

Response – we added the discussion on page 14.

4) *It is surprising that the transient transfection of LAPC4 cells, which typically have a low transfection efficiency, with a GPI expression plasmid was sufficient to promote a significant level of therapy resistance in a population of cells. Controls should be included to demonstrate transfection efficiency and confirm GPI overexpression.*

Response – we have added additional experimental results to show the overexpression of GPI in LAPC4 cells, including the western blots of GPI overexpression (Supplemental figure 8B), and the enzymatic activity of the overexpressed GPI (Supplemental figure 8C, and figure 6F).

5) *GPI levels or activity should be assessed in the tumor samples from Figure 1G to support the model.*

Response – we used qRT-PCR to extensively evaluate the expressions of GPI, ENO1 and TMPRSS2 in tumor samples from figure 1G. The results are shown in figure 7A-7D / page 11.

6) *Given that the functional effects of GPI is a major point of this study and concerns regarding the off-target effects of chemical siRNAs, the experiments described in Figures 5D-G and 6B should be confirmed using multiple siRNAs, an add-back approach or the CRISPR GPI knockout line created and used in Figure 6G.*

Response – we performed the GPI adback experiment to a) confirm the GPI-siRNA, and b) confirm the efficacy of GPI overexpression (Supplemental figure 8C). Also, we show that the CRISPR-GPI knockout had similar experimental results to GPI-siRNA (Supplemental figure 12 / page 11 in text).

7) While potentially beyond the scope of this study, evidence of increased glycolysis in the resistant tumors from any of the animal studies would greatly strengthen the data.

Response – we thank the reviewer for recognizing the scope of our current study. In figure 7F, the enzalutamide-resistant LAPC4-AdtHs xenograft had increased sensitivity to 2DG + enzalutamide treatment. Since 2DG is a glycolysis inhibitor, this suggested to us that there was an increase of glycolysis and it was essential to the enzalutamide resistance in this in vivo model.

Minor points:

1) JQ1 is not a direct AR inhibitor. Rather it indirectly targets AR via associated BRD4.

Response – We have removed the sentence of JQ1 as a direct AR inhibitor, and describe it as an inhibitor of AR-associated BRD4 on page 8.

2) It is not clear why repeated cycles of ADT were needed compared to continuous ADT.

Response – Continuous hypoxia in cell culture conditions (over 120 hours) inevitably leads to cell death to cells without prior shorter-hypoxic selection/preconditioning step. Thus, we had to use multiple cycle of treatment for all of our chronic ADT. We have removed the word “repeated”, and use the word “chronic” to be consistent with the chronic in vitro treatment model.

3) Comment: It is worth noting that subcutaneous injection sites are quite hypoxic and hence, without stating so, the authors chose an ideal xenograft model site for their studies. While certainly beyond the scope of this study, it would be interesting to see if similar effects would be observed at other, less hypoxic sites.

Response – Thanks for recognizing the scope of our in vivo experiments. Indeed, we are also interested in conducting in vivo experiments in additional sites for future studies.

4) It is premature to conclude that ATP production and energy homeostasis are the main reasons why glycolysis would be driving disease progression in this context. This was never formally tested and there are many other important aspects of glycolysis that contribute towards tumor growth and cancer spreading.

Response – We fully agree with reviewer’s opinion. Thus, we have removed the discussion regarding ATP production as the main reason of resistance. Instead, we discuss more in term that the therapy-induced switch maintains the flux of glucose metabolism on page 14.

5) Catalog numbers for the metabolic kits should be provided because it is not clear how they assessed F6P levels. The stated company does not appear to offer an assay kit to quantitate this metabolite, which is often difficult to distinguish from G6P.

Response – all catalog numbers are provided in the material and methods section (page 21). Specifically, the catalogue # for F6P measurement is BioVision #K689.

6) Where repeated comparisons are made, repeated t tests are likely inadequate. Rather, ANOVAs with appropriate post hoc tests should be used.

Response – as stated throughout the figure legends, in addition to the t-test, we have also performed repeated measures ANOVA with Bonferroni adjustment for statistical analyses involving more than two groups of samples.

7) Some of the stat comparison bars are misaligned in Figure 4.

Response – the stat comparison bars have been re aligned.

8) Was GPI included in any of the highlighted GSEA pathways?

Response – Yes, as shown in Supplemental Figure 3C (page 39), GPI is included in one of the highlighted GSEA pathways – mTORC1.

9) *Typo in Supp Fig 13B y axis.*

Response – Typo has been corrected.

Reviewer #2:

The authors have taken a novel approach to identify factors which can contribute to ADT-resistance under conditions of hypoxia. This has evolved the adaptive selection of cell-lines through repeated exposure to hypoxia and ADT. Of the prognostic genes identified in this study the authors have focused on GPI and its contribution to sustaining glycolysis under conditions of hypoxia once androgen-mediated repression has been removed. The study is provocative and carefully undertaken with respect to GPI.

*The manuscript will need some **minor revisions** prior to publication including the full deposition of microarray data in a suitable repository.*

Response – we have submitted the microarray data to the Gene Expression Omnibus (GEO).

The authors have used the TCGA dataset to associate the prognostic value of genes that are induced by hypoxia and repressed by androgen but highly expressed under conditions of ADT and hypoxia. They have done so through single-gene analysis. Are all of these prognostic genes differentially expressed in the same subset of TCGA cases?

Response – In the new Figure 2G / page 5, we are showing the prognostic value of the 10-gene prognostic geneset as a group in the TCGA dataset. In the new Supplemental Figure 4B / page 41, we are showing the frequency of mRNA alteration for the 10-gene geneset. As it appears that multiple alteration may occur to the same patient.

Which pathways are dysregulated in cases identified by these genes?

Response – In the new Supplemental Figure 4B / page 41, we provide the pathways involving the dysregulated genes.

How does the GSEA analysis undertaken in supplementary figure 3 and enrichments shown here map back to these genes?

Response – In the new Supplemental Figure 3C (page 39), we provide the list of genes involved in the differential GSEA analysis. The genes that we also identified as important through FDR-t-test (map back) are highlighted and underlined.

Which genes account for the enrichments attributed to biology in this analysis - eg. Androgen response or estrogen response. These should be provided in tabular form to provide extra insights.

Response – As shown in Supplemental Figure 3C and 3D (page 39-40), the genes that were responsible for the GSEA pathway enrichment are all tabulated.

Finally, of the genes that have been reported to be prognostic in the TCGA data, including GPI, are any also known to be regulated by other poor prognosis mutant drivers of prostate cancer - eg. TP53, PTEN etc. This extra information will significantly enhance the impact of this study.

Response – As shown in Supplemental Figure 4B (page 40), the 10-gene geneset that may have prognostic value based on TCGA dataset is involved in different cancer hallmark pathways, including the poor prognosis mutant drivers of prostate cancer such as P53, mTORC1.

Reviewers' Comments:

Reviewer #1:

Remarks to the Author:

The authors did an excellent job addressing this reviewer's comments and therefore strengthening this interesting study.

Reviewer #2:

Remarks to the Author:

The authors have addressed my comments. Content for the paper to proceed.